# Fresh and aged aerosols emitted from biomass burning in northern Australia

Andelija Milic[1], Marc D. Mallet[1], Luke T. Cravigan[1], Joel Alroe[1], Zoran D. Ristovski[1], Paul Selleck[2], Sarah J. Lawson[2], Jason Ward[2], Maximilien J. Desservettaz[3], Clare Paton-Walsh[3], Leah R. Williams[4], Melita D. Keywood[2], Branka Miljevic[1]

[1]International Laboratory for Air Quality and Health, Queensland University of Technology, Brisbane, Queensland, 4000, Australia
[2]CSIRO Oceans and Atmosphere, Aspendale, Victoria, 3195, Australia
[3]Centre for Atmospheric Chemistry, University of Wollongong, Wollongong, New South Wales, 2522, Australia
[4]Aerodyne Research, Inc., Billerica, Massachusetts, 01821, USA

*Correspondence to*: Branka Miljevic (b.miljevic@qut.edu.au)

**Abstract.** There is a lack of knowledge of how biomass burning aerosols in the tropics age, including those in the fire-prone Northern Territory in Australia. This paper reports chemical characterization of fresh and aged aerosols monitored during the one month long SAFIRED (Savannah Fires in the Early Dry Season) field study, with an emphasis on chemical signature and aging of organic aerosols. The campaign took place in June 2014 during the early dry season when the surface measurement site, the Australian Tropical Atmospheric Research Station (ATARS), located in the Northern Territory, was heavily influenced by thousands of wild and prescribed bushfires. ATARS was equipped with a wide suite of instrumentation for gaseous and aerosol characterization. A compact time-of-flight aerosol mass spectrometer was deployed to monitor aerosol chemical composition. Approximately 90 % of submicron non-refractory mass was composed of organic material. Ozone enhancement in biomass burning plumes indicated increased air mass photochemistry and increased organic aerosol with the aging parameter (f44) suggested secondary organic aerosol formation. Diversity of biomass burning emissions was illustrated through variability in chemical signature (e.g. wide range in f44, from 0.06 to 0.18) for five intense fire events. The background particulate loading was characterized using Positive Matrix Factorization (PMF). A PMF-resolved BBOA (biomass burning organic aerosol) factor comprised 24 % of the submicron non-refractory organic aerosol mass, confirming the significance of fire sources. A dominant PMF factor, OOA (oxygenated organic aerosol), made up 47 % of sampled aerosol fraction, illustrating the importance of aerosol aging in the Northern Territory. Biogenic isoprene-derived organic aerosol factor was the third significant fraction of the background aerosol (28 %).

**Keywords.** Biomass burning, savannah fires, organic aerosol, aerosol aging

## 1 Introduction

Tropical regions are some of the most fire-prone areas in the world (Crutzen and Andreae, 1990). As one of them, Northern Australia is observed to be a significant contributor to the global biomass burning (BB) budget (Russell-Smith et al., 2007). Fire emissions are made up of a variety of gaseous and particle-phase species, with the particle-phase consisting primarily of carbonaceous particles, including organic carbon (OC) and elemental carbon (EC) (Bond et al., 2004;Hallquist et al., 2009;Formenti et al., 2003;Crutzen and Andreae, 1990). Open fires are the largest contributors to carbonaceous emissions; approximately 70 % and 40 % of the global annual emissions of OC and EC are formed in open fire events, respectively (Bond et al., 2004). Annually, 60-75 % of Australia's carbon emissions are attributed to BB emissions in tropical savannahs (Hurst et al., 1994). Organic carbon and EC affect the Earth's radiation balance by scattering and absorbing solar radiation, respectively (Penner et al., 1998;Haywood and Boucher, 2000). O'Brien and Mitchell (2003) suggested that atmospheric heating caused by UV absorbing carbonaceous aerosol, related to BB events in the Northern Territory during a dry season, can be significant and will likely have an influence on local (and possibly even global) climate. Carbonaceous species can also serve as condensation nuclei in cloud formation processes (Roberts et al., 2002). It is therefore important to build a comprehensive knowledge of BB emissions as they play a key role in the climate.

The characterization, processing and estimation of impact of BB emissions is a complex and challenging issue due to a wide range of burning material and combustion conditions (flaming/smouldering), as well as complex atmospheric physics and chemistry that BB plumes undergo once emitted (Reid et al., 2005;Weimer et al., 2008). Besides the characterization of fresh BB emissions, it is important to address their evolution in the atmosphere since aging of the BB emissions will alter their properties and thus how they interact with the climate system. Fresh BB emissions are those released directly from BB sources, while processed emissions refer to fresh particle or gaseous species transformed in the atmosphere through photochemical processing (aging). Organic aerosols (OA) can be differentiated as directly emitted primary organic aerosols (POA), aged primary organic aerosols (aged POA) and secondary organic aerosols (SOA) that form via condensation of lower volatility photo-oxidation products. A large fraction of BB-related POA is observed to be semivolatile (Grieshop et al., 2009) and can therefore change upon dilution with background air by evaporation of its content into the gas phase and be exposed to further transformations in the atmosphere.

Processing of BB emissions in the ambient atmosphere and in laboratory experiments is characterized by increases in the oxygenated fraction of OA (OOA) and degradation of species that are indicators of BB origin (Cubison et al., 2011;DeCarlo et al., 2008;Capes et al., 2008). However, there is no agreement on whether the aging of BB emissions leads to additional OA production. Some recent field and laboratory studies investigating the evolution of BB plumes have shown that OA mass increases with aging (DeCarlo et al., 2008;Yokelson et al., 2009;Heringa et al., 2011), while others have found no significant

change or a decrease in OA with atmospheric processing (Akagi et al., 2012;Brito et al., 2014;Cubison et al., 2011;Capes et al., 2008). In addition to SOA formation, BB plumes promote the production of another important secondary pollutant, tropospheric ozone (Real et al., 2007). An excess of nitrogen oxides ($NO_X$), non-methane organic compounds (NMOCs) and carbon monoxide (CO) in the atmosphere from fire emissions promote additional pathways for ozone production (Parrington et al., 2013). Ozone enhancement in BB emissions has been extensively used as a proxy for air mass photochemical activity (Hobbs et al., 2003;Jaffe and Wigder, 2012;Yokelson et al., 2009;Akagi et al., 2012).

An extensive area of the Northern Territory of Australia is burned each year during the dry season (May-October) and the region is strongly affected by BB emissions. Considering that this area is a globally significant source of BB aerosol emissions (Mitchell et al., 2013), and that there have been a limited number of studies on aerosol characterization and aging, there is a need for more research. Information about the composition and degree of aging of OA that makes up the majority of emitted species generated in these fires will facilitate the estimate of the influence of BB in the Northern Territory. This will also contribute to a better understanding of fire emissions in tropical regions. In order to address these issues, the one month long SAFIRED (Savannah Fires in the Early Dry Season) field study was conducted at the Australian Tropical Atmospheric Research Station (ATARS) during a frequent burning period (late May/June 2014). This publication presents insights into fresh and aged aerosols emitted during the SAFIRED, while a detailed description of the campaign can be found in Mallet et al. (2016).

## 2 Measurement site and period

The Australian Tropical Atmospheric Research Station (ATARS) is situated on the Gunn Point peninsula in the Northern Territory (12°14'56.6" S, 131°02'40.8" E). The northern Australian tropical land mass is mostly covered by savannah biomes including scattered eucalypt trees, shrubs and grasses (Lacey et al., 1982). The sampling site can be described as rural background due to its remote location, with the closest populated centre, the city of Darwin, approximately 80 km south west of the ATARS. During the measurement period, the dominant sources of gaseous and aerosol species were wild and prescribed fires. Apart from planned burns, anthropogenic inputs to this region can be considered negligible (Bowman et al., 2007) and observations have highlighted the importance of biogenic sources for this area (Allen et al., 2008). The sampling site can also be impacted by marine aerosols (Bouya et al., 2010); however, fires are the major source of aerosols during the dry season.

The SAFIRED campaign was conducted from the 29[th] of May until the 30[th] of June, 2014 at the beginning of the dry season. The early dry period is characterized by natural fires as well as prescribed burns conducted to reduce the extent, intensity and frequency of wildfires in the late dry season (October-November). Thousands of fires in the Northern Territory during the campaign were detected by MODIS and VIIRS satellite sensors (Mallet et al., 2016). Fire emissions sources ranged from approximately two to up to hundreds of kilometres distant from the sampling station.

The early dry season was characterized by dry weather conditions (average relative humidity of 67 ± 23 %) and warm days with an average daily and nightly temperatures of 27 ± 5 °C (up to 34 °C) and 19 ± 4 °C (with a minimum of 10 °C), respectively. There were no days of rainfall. Wind direction was predominantly southeast, suggesting that the sampled air masses had mainly passed over land affected by fire emissions. However, on some days (e.g. between 3[rd] and 6[th] of June) in the afternoon hours northeast wind direction was dominant, directing the air masses from land to pass over the ocean before being detected at ATARS. Moreover, an oceanic influence was observed between 19[th] and 22[nd] of June (Mallet et al., 2016).

## 2.1 Instrumentation and method

Ground level characterization was performed using a wide range of instrumentation for gas- and particle-phase measurements. Detailed description of all instruments used in SAFIRED is given in Mallet et al. (2016). The focus of this study was on the aerosol chemical composition and aging using the instrumentation listed in the following paragraphs.

### 2.1.1 Compact Time-of-Flight Aerosol Mass Spectrometer (cToF-AMS)

A compact time-of-flight aerosol mass spectrometer (cToF-AMS or AMS, Aerodyne Research, Inc.) was deployed to monitor the chemical composition of non-refractory submicron ($PM_1$) aerosol. Monitored species were submicron particulate organic and inorganic compounds (sulfate, ammonium, nitrates and chlorides). Details about instrument operation during this campaign can be found in Mallet et al. (2016).

Unit mass resolution (UMR) and high resolution (HR) AMS data analysis was performed in Wavemetrics Igor Pro software (version 6.36) using the standard AMS Analysis Toolkits, Squirrel (Version 1.56D) and PIKA (Version 1.16), respectively. Data collection with a filter at the instrument inlet was used to correct for contributions from air in the fragmentation table (Aiken et al., 2008). Measured time-resolved gas-phase $CO_2$ concentrations were applied instead of the fragmentation table default value. The AMS collection efficiency was determined using the calculations provided within the PIKA Toolkit. The composition dependent collection efficiency panel enables an estimate of the collection efficiency based on ammonium nitrate, organic content, aerosol acidity and relative humidity (Middlebrook et al., 2012). The time-resolved collection efficiency (with an average value of 0.61) was applied to the entire dataset. High OA loadings during the campaign caused interferences in the detection of sulfate in the UMR analysis. Significant improvement was made in distinguishing sulfate fragments from organic fragments at the same m/z by performing HR peak fits in PIKA (Sect. S1 in supplementary information). Therefore, HR peak fitting data (PIKA) were chosen for further analysis.

### AMS fragments analysis

Organic aerosols measured by the AMS encompassed aerosols that were processed in the atmosphere for different periods of time and included both POA and SOA. As such, a tool was needed to distinguish BB aerosol from other sources and fresh from processed BB aerosol. An extensively employed fragments analysis using the AMS-extracted parameters f43 (ions

$C_3H_7^+$ and $C_2H_3O^+$), f44 ($CO_2^+$) and f60 ($C_2H_4O_2^+$) was applied here. The ratio of an integrated organic signal, e.g. m/z 43, to total organic signal is referred to as f43. Parameters f43 and f44 were used to estimate the level of OA processing, as they illustrate OA fractions of different aging degree. While f43 is mainly associated with hydrocarbon-like and semivolatile OA fractions (fragments $C_3H_7^+$ and $C_2H_3O^+$, respectively) (Jimenez et al., 2009;Lanz et al., 2007;Zhang et al., 2007;Heringa et al., 2011), f44 is assigned to highly oxygenated OA species (fragment $CO_2^+$) (Aiken et al., 2007;DeCarlo et al., 2010;Alfarra et al., 2007). The AMS parameter f60 and accompanying f73 ($C_3H_5O_2^+$) are widely used as BB emission signatures as they are directly related to levoglucosan-like species, which are a substantial fraction of organics emitted in pyrolysis of cellulose (Alfarra et al., 2007;Simoneit et al., 1999). The f60 can be applied as a valuable BB marker for time periods of up to one day (Cubison et al., 2011;Bougiatioti et al., 2014). Ambient aerosols characterized by an f60 value higher than 0.003 are considered to be influenced by BB emissions (Cubison et al., 2011). A graphical approach of comparing f44 vs. f60 introduced by Cubison et al. (2011) was used in this study to estimate the degree of aging of BB emissions.

**Positive Matrix Factorization (PMF)**

Positive Matrix Factorization (PMF) analysis (Paatero and Tapper, 1994) using the PMF evaluation tool (Ulbrich et al., 2009) was used in source apportionment of the AMS OA data. PMF splits the OA spectrum into different factors that can be related to specific sources. (Paatero, 1999;Paatero and Hopke, 2009). Various factors have been identified by PMF analysis of AMS OA data, with the most common being hydrocarbon-like OA (HOA) and oxygenated OA, that can sometimes be further apportioned to low-volatility oxygenated OA (LV-OOA) and semivolatile oxygenated OA (SV-OOA) (Ulbrich et al., 2009;Lanz et al., 2007;Ng et al., 2010;Jimenez et al., 2009). Other factors include environment specific factors such as biomass burning OA (BBOA) and cooking-related OA (COA). The solution space in this study was investigated by varying the number of factors and starting conditions (seeds, 0 to 50 in increments of 1) and applying rotational parameters (fpeaks, -1 to 1 in increments of 0.1).

**2.1.2 Beta plus particle measurement system (BAM)**

$PM_1$ mass was measured and collected onto pre baked (600 °C) quartz 47 mm filters (Pall Tissuquatz p/n 7202) using a Beta plus particle measurement system (BAM, Teledyne API Model 602), every 12 hours. All of the species measured on BAM filters were blank corrected. Particles collected on filters were further analysed for anhydrous sugars, including levoglucosan, by high performance anion exchange chromatography with pulsed amperometric detection and for water soluble ions including potassium ($K^+$), nitrates ($NO_3^-$), sulfates ($SO_4^{2-}$), ammonium ($NH_4^+$) and chlorides ($Cl^-$) by ion suppressed chromatography. Levoglucosan and non-sea salt $K^+$ were used as metrics for BB emissions. BAM $PM_1$ filters were analysed by a Thermal-Optical Carbon Analyser (DRI Model 2001A) using the IMPROVE-A temperature protocol (Chow et al., 2007) to determine contributions of EC and OC. Water soluble inorganic ions and OC were also used for comparison with AMS-resolved species.

### 2.1.3 Fourier Transform Infrared Spectrometer (FTIR)

The Spectronus trace gas and isotope Fourier Transform Infrared Spectrometer (FTIR) analyser, built by the Centre of Atmospheric Chemistry at the University of Wollongong, was deployed to monitor gaseous species including carbon monoxide (CO) and carbon dioxide ($CO_2$) (Griffith et al., 2012;Griffith, 1996). Details about the instrument operation can be found elsewhere (Desservettaz et al., 2016). In order to include dilution effects, OA and ozone data are presented relative to CO. CO is an important BB product that can remain in the atmosphere for a relatively long time period (one to two months depending on the environment) without observed decay or interactions with oxidative agents (Wang and Prinn, 1999;DeCarlo et al., 2010;Edwards et al., 2006). CO has therefore been employed as a long-lived, conservative tracer in many studies (DeCarlo et al., 2010;Kleinman et al., 2008;Brito et al., 2014;Yokelson et al., 2009;Akagi et al., 2012). Gas-phase $CO_2$ data, as previously mentioned, were applied in the AMS fragmentation table adjustment. CO and $CO_2$ were also used in modified combustion efficiency (MCE) calculations, with details presented in Desservettaz et al. (2016). MCE refers to the ratio of $\Delta CO_2$ (measured value relative to background value) to the sum of $\Delta CO_2$ and $\Delta CO$ (Ward and Radke, 1993). The MCE parameter was used to distinguish smouldering (usually lower than 0.9) from flaming (usually higher than 0.9) fires. The MCE value can also indicate the burning material.

### 2.1.4 Scanning Mobility Particle Sizer (SMPS)

Aerosol size and number concentration were monitored by a Scanning Mobility Particle Sizer (SMPS, a TSI 3071 long-column electrostatic classifier coupled to a TSI 3772 Condensation Particle Counter). The SMPS measured the particle size distribution from 14 nm up to 670 nm every 5 minutes. Changes in the size distribution due to aging were investigated.

### 2.1.5 Proton Transfer Reaction-Mass Spectrometer (PTR-MS)

A high sensitivity Proton Transfer Reaction-Mass Spectrometer (PTR-MS, Ionicon Analytik) with a quadrupole mass spectrometer and an $H_3O^+$ ion source was employed to measure non-methane organic compounds (NMOCs) that include non-methane hydrocarbons and oxygenated volatile organic compounds. These species are important in the formation of SOA and ozone. Isoprene and monoterpenes make up a dominant fraction of global gas-phase NMOCs, contributing significantly to the production of SOA (Guenther et al., 2012). Tropical regions can contribute up to 80 % of global annual isoprene emissions due to warm weather conditions (Guenther, 2006). He et al. (2000) investigated NMOCs emitted from eucalyptus trees, which make up 95 % of all tree species in Australia. Isoprene accounts for 64-100 % of NMOCs emitted from different eucalyptus species. The isoprene/furan fraction (signal at m/z 69) of measured NMOCs is used here in the analysis of isoprene-derived OA. Other studies suggest that the contribution of isoprene to m/z 69 prevails in non-BB periods while the furan contribution is more significant in BB plumes (Warneke et al., 2011). In order to distinguish furan and isoprene contribution over the sampling period, a gas chromatography-mass spectrometry method was used.

**2.1.6 Ozone analyser**

Ozone concentration was monitored by an Ultraviolet Photometric Ozone analyser (Model 49i, Thermo Scientific). Ozone enhancement in BB emissions has been used as a proxy for air mass photochemical activity.

**2.1.7 Fire data**

Information about the location and duration of fires was obtained from the Sentinel Hotspots system. Hotspots data used for this campaign were derived from the MODIS (Moderate Resolution Imaging Spectroradiometer) sensor (Terra and Aqua satellites) and VIIRS (Visible Infrared Imaging Radiometer Suite) sensor (Suomi NPP satellites). Only the hotspots with confidence level higher than 50 % were considered in the analysis. Hotspots are detected by satellite once a day which limited the fire analysis to events that occurred between approximately 11 am and 3 pm. During the SAFIRED campaign thousands of fires were detected in the Northern Territory. Distance-resolved fire frequencies are presented in Mallet et al. (2016). Moreover intense fires characterized by plumes detected without considerable influence from other fire sources, were extracted from the whole dataset (Desservettaz et al., 2016). Five events that were categorised as single, intense fires are analysed here. Air mass backtrajectories were computed using the NOAA/ARL HYSPLIT (Hybrid Single-Particle Lagrangian Integrated Trajectory) model.

**3. Results and Discussion**

**3.1 Aerosol chemical characterization**

$PM_1$ ambient aerosols sampled during the SAFIRED campaign were dominated by an organic fraction. Organic carbon made up 72 % and EC 15 % of the measured $PM_1$ on the BAM filters (Fig. 1a). Data from the AMS (Fig. 1b and 1c), indicate that organics made up approximately 90 % of submicron non-refractory mass with an average concentration of 11.1 µg m$^{-3}$ and concentrations of up to 350 µg m$^{-3}$ during intense and nearby BB events. The organic mass (OM) sampled by the BAM was converted from OC using the conversion factor of 1.4, which was determined in PIKA. This value is within the span of OM/OC conversion factors for the biomass burning aerosol suggested by Reid et al. (2005). The BAM OM and mass concentration of organics sampled by the AMS are in good agreement (R value of 0.94), with slightly higher concentrations measured by the BAM (Fig. S2c) probably due to the lower collection efficiency of the AMS above 700 nm. Organic mass has been observed to be the dominant fraction of $PM_1$ ambient aerosols during the early dry season in the Northern Territory (Carr et al., 2005). The organic fraction was also dominant in other studies strongly impacted by BB emissions (Brito et al., 2014;Capes et al., 2008). In this study, the remaining submicron non-refractory mass was made up of inorganics including sulfates (4.2 %), ammonium (2.8 %), nitrates (1.5 %) and chlorides (1.3 %), with average concentrations of $0.51 \pm 0.32$ µg m$^{-3}$, $0.35 \pm 0.68$ µg m$^{-3}$, $0.19 \pm 0.45$ µg m$^{-3}$ and $0.17 \pm 1.28$ µg m$^{-3}$, respectively.

The sampling site was constantly impacted by fire emissions with periods of heavy BBs characterized by high aerosol and gas phase concentrations, for instance CO reaching up to ~$10^4$ ppb and organics up to 350 µg m$^{-3}$ (Fig. 2). The most intense BB episodes were on the 30th of May, between the 7th and 11th of June, and on the 25th and 26th of June (Fig. 1c) (Mallet et al., 2016). During intense fire periods, organics, CO and particle number concentration showed correlated increases (Fig. 2).

Moreover, looking at Sentinel Hot spots during these periods, hot spots were detected within 20 km from the ATARS. Based on this, the dataset was separated into periods of "close BB" (corresponding to high organics, CO and particle number concentration signals and close events (< 20 km)) and "distant BB" (corresponding to less intense organics, CO and particle number concentration signals and distant events (> 20 km)). It is important to emphasize that all periods during the measurement that have not been included in close BB periods have been assigned to distant BB periods, as ATARS was

constantly influenced by BBs. The selection does not mean that emissions from distant fires were not present during the close BB periods but that the influence of fires near the measurement station during these periods was dominant. Nine intense BB events were identified from close BB periods as single source emissions (Desservettaz et al., 2016). Five of the nine events (Fig. 2) were analysed here (Sect. 3.2.4), due to the instrument not sampling during the remaining events. Most of the events occurred in the afternoon/night time (Table 1).

Coupled with elevated organic concentrations, AMS-measured inorganics also increased during close BB periods (Fig. 1c). A high AMS signals for all monitored species corresponded to close BB periods. During SAFIRED, high correlation was observed between organics and ammonium species (R of 0.92), with better agreement during the close BB (R of 0.92) than in the distant BB periods (R of 0.73) (Table S1), suggesting that these ammonium species originated from fire emissions. Moreover, plume emissions included high chloride concentrations (up to approximately 50 µg m$^{-3}$ during close BB periods).

Increased chloride concentration during the close BB periods was also indicated by the BAM data (Fig S3a). This is not surprising since the dominant fine particle inorganic fraction emitted in flaming savannah fires has been shown to be composed of KCl (potassium chloride) and $NH_4Cl$ (ammonium chloride) compounds and/or their mixtures (Liu et al., 2000). There was a better correlation between organics and chlorides for close BB periods (R of 0.67), than for distant BB periods (R of 0.47). This can be explained by the depletion of chloride with transport and aging of BB plumes (Li et al., 2003;Li et

al., 2010;Liu et al., 2000). Correlations for nitrate and sulfate with organics show similar patterns in this work regardless of the BB emissions influence (R of 0.72 and 0.77 for nitrate and R of 0.49 and 0.48 for sulfate for close and distant periods, respectively). AMS inorganic species were compared to soluble ions concentrations measured on the filter samples collected using the BAM. There was a strong correlation between all compared species (R values of 0.85, 0.68 and 0.81 for sulfates, ammonium and nitrate, respectively) except in the case of chloride (R of 0.18) (Fig. S2). $Mg^{2+}/Na^+$ ratio values for the filters

collected during the high oceanic influence (between 3rd and 6th and 19th and 22nd of June) were close to the sea salt $Mg^{2+}/Na^+$ ratio of 0.12 (Fig. S3b). At the same time low terrestrial fetch (low radon concentration) was observed (Mallet et al., 2016). Moreover, the chlorides collected on filters were prominent during the period of oceanic influence (Fig. S3a). This

suggests that a significant portion of chlorides detected on the BAM filters was of a sea salt origin, which is refractory and therefore not well measured by the AMS, thereby explaining the poor chloride correlation.

## 3.2 Biomass burning aerosols and aging

In general, BB-related particulate matter can be easily distinguished from other aerosol sources by chemical composition.
Levoglucosan is a common molecular tracer of plume emissions (Simoneit et al., 1999) and has been extensively used as a BB chemical signature (Jordan et al., 2006;Simoneit, 2002). Moreover, the AMS organic signal intensity (fragment $C_2H_4O_2^+$) (Org 60) is directly correlated to the concentration of levoglucosan-like species (Alfarra et al., 2007;Simoneit et al., 1999) and can therefore be applied as a valuable BB marker. Fire sources also contribute to high potassium levels (Li et al., 2003). Figure 3 illustrates the time series of Org 60 obtained from the AMS and levoglucosan and soluble non-sea salt potassium
sampled by the BAM. Prominent signals of these BB tracers were present during the observed close BB periods, which confirm that the fire source of the detected aerosols is BB.

In order to discriminate BB aerosol from other sources and estimate the degree of BB aerosol aging (processing), a graphical method introduced by Cubison et al. (2011) has been applied here (Fig. 4). The majority of data points (94 % of the dataset) had f60 values above the background limit of 0.003, indicated with a vertical dashed lines in Fig. 4. Thus, the detected OA
during SAFIRED can be attributed mostly to BB aerosol.

Chemical aging of BB-related OA typically leads to f60 depletion and increased f44 (Grieshop et al., 2009;Cubison et al., 2011;Ortega et al., 2013;Ng et al., 2010;Zhao et al., 2014;Capes et al., 2008;Jolleys et al., 2015). In general, this trend was observed in this study for distant fires, with the data trending towards the top left corner of the diagram (Fig. 4b). Apart from aerosol processing, changes in f44 and f60 can be attributed to burning conditions and/or materials. The variability observed
in f44 vs. f60 for close BB events (Fig. 4a) can reflect BB plumes generated during different burning conditions but also different atmospheric processing of BB masses. On average distant BB plumes were characterized by lower fire tracer f60 compared to the close BB emissions. Furthermore, distant fire plumes indicated a higher portion of oxygenated compounds (i.e., higher f44 values), relative to close plumes. The maximum f44 value ranged between 0.20 and 0.23 in this study, which agrees well with previously observed f44 values for oxidized BB emissions in ambient measurements (Cubison et al., 2011).
The f44 parameter increased to values characteristic of LV-OOA in the distant BB and yet f60 values were substantially above the background value of 0.003. This confirms that levoglucosan-like species carried by the BB plume did not degrade to background levels even as oxidized species were formed. Thus, f60 is a reasonable marker of distant BB in this study.

All data fall into the f44 vs. f43 triangular plot range for ambient data, introduced by Ng et al. (2010) (Fig. S4). Ng et al. (2010) observed that typical ambient OA data slopes from the bottom right to the top left of the f44 vs. f43 plot, and that this
trend can be attributed to photochemical processing of ambient air masses. A similar pattern was observed in this study for distant periods (Fig. S4b). A wide range in f44 can be attributed to different processed (aged) BB-related OA, but different

burning material and/or conditions cannot be excluded. The observed evolution trend in f44 vs f43 for distant fires can be also influenced by mixing between the plumes. In contrast the f43 values for close BB (Fig. S4a) are located in a narrow f43 range from 0.05 to 0.08 and do not change considerably with increasing f44. This can be a result of insignificant atmospheric processing in case of close fires. The wide range in f44 can be attributed to difference in burning conditions for close BB.

One more factor that can influence f44 vs f60 trend for both close and distant fire is dilution effect.

### 3.2.1 Ozone formation

The emissions from BB sources promote ozone production in the troposphere by increasing the concentrations of key ozone-forming precursors (NMOCs, $NO_X$ and CO). Air mass photochemical activity can be estimated through the observation of changes in ozone concentration with aging, particularly the change in the $\Delta O_3/\Delta CO$ ratio, where $\Delta CO$ and $\Delta O_3$ refer to

enhancements above background concentrations. $\Delta CO$ accounts for dilution, as it is a relatively long lived atmospheric species. The minimum CO value of 80 ppb measured during June at the ATARS site was used as the background concentration level. Similarly a minimum ozone concentration of 10 ppb was considered to be the background level in these calculations. On average, the $\Delta O_3/\Delta CO$ ratio increases with f44 and decreases with f60, indicating increased photochemical processing of OA in plumes with atmospheric aging and ozone production (Fig. 4c and Fig. 4d).

$\Delta O_3/\Delta CO$ values vary between -0.1 and 0.9 which is within the observed range in other studies (Jaffe and Wigder, 2012). A negative ratio may indicate fresh BB emissions where ozone was removed by atmospheric reaction with nitric oxide (NO) emitted in high amounts from fire sources, as suggested by Yokelson et al. (2003). Close fire emissions were characterized by significantly lower ozone enrichments (average $\Delta O_3/\Delta CO$ of 0.15) compared to plumes detected from the distant BB emissions (average $\Delta O_3/\Delta CO$ of 0.31) which represent higher photochemical activity within more processed air masses.

These values agree with observed $\Delta O_3/\Delta CO$ ratio values for plumes aged for less than 1-2 days (average $\Delta O_3/\Delta CO$ of 0.14) and for 2-5 days (average $\Delta O_3/\Delta CO$ of 0.35) in tropical/subtropical regions (Jaffe and Wigder, 2012). The $\Delta O_3/\Delta CO$ enrichments for close BB period indicate that aging of close emissions cannot be excluded. These significant ozone enrichments in BB plumes illustrate high $NO_X$ and NMOC loadings emitted from fire sources and photochemically active air masses.

### 3.2.2 Secondary organic aerosol (SOA) formation

Since increased photochemical activity was identified in BB air masses, the change in $\Delta OA/\Delta CO$ ratio was investigated in order to determine whether additional OA was produced in the BB plumes (Fig. 4e and Fig. 4f). The lowest OA concentration observed during the campaign of 0.09 $\mu g\ m^{-3}$ was taken as a background value in these calculations. Figures show no particular trend in $\Delta OA/\Delta CO$ ratio with f44. The $\Delta OA/\Delta CO$ ratio also remains quite constant despite increases in

f44 (Fig. S5). In addition, diurnal patterns of the f44, $\Delta O_3/\Delta CO$ and $\Delta OA/\Delta CO$ ratios were investigated, for close and distant BB periods separately (Fig. 5a and Fig. 5b, respectively). The parameter f44 was used to indicate the level of oxygenation

that can be caused by the photochemical changes and $\Delta O_3/\Delta CO$ as a parameter of photochemical activity. It must be noted that simply examining $\Delta OA/\Delta CO$ vs time of day is a simplified approach which does not fully take into account the total photochemical history of the air mass. There was an increase in $\Delta OA/\Delta CO$ ratio with an increase in f44 and $\Delta O_3/\Delta CO$ in the late morning and early afternoon for close and for distant BB periods. This is likely due to condensation of organics onto

pre-existing particles. The decrease in the $\Delta OA/\Delta CO$ ratio later in the afternoon could indicate OA loss due to fragmentation and subsequent evaporation from the particulate phase. As suggested by Kroll et al. (2009), OA loss can reflect dominance of the fragmentation pathways in the formation of more oxidized OA. Therefore it is suggested that increased photochemical activity of BB-influenced air masses, illustrated by increase in $\Delta O_3/\Delta CO$ and likely due to oxygenation of OA (f44 increase), was accompanied by an increase in $\Delta OA/\Delta CO$ ratio, indicating SOA formation. Moreover, the decrease in $\Delta OA/\Delta CO$ ratio

later in the day can be a result of fragmentation and subsequent evaporation.

### 3.2.3 f44-resolved size distribution

Atmospheric aging of plume particles increases particle diameter due to gas to particle transfer of organic and inorganic gaseous species (Martins et al., 1998). In order to estimate whether the aging of BB masses during SAFIRED influenced particle size, average SMPS size distributions and AMS size distributions for organics, both categorised based on different

f44 ranges, were examined (Fig. 6). It is important to emphasise that SMPS uses electrical mobility diameter, while AMS uses vacuum aerodynamic diameter. The close and distant BB periods were analysed separately. The f44 values were classified into four groups that represent different aging stages (0.05<f44<0.1, 0.1<f44<0.15, 0.15<f44<0.2, 0.2<f44<0.25). The first f44 bin (0.05<f44<0.1) was not considered in case of the distant BB periods, as only a few data points were in this range (Fig. 4). The same was done for the highest f44 bin (0.2<f44<0.25) for close BB periods. According to SMPS data, the

average particle mode varied between 101 – 113 nm and 104 – 106 nm for close and distant BB periods, respectively. The average mode for organics showed larger sizes and varied between 259 – 293 nm and 293 – 305 nm for close and distant BB periods, respectively. Increased f44 was accompanied by a reduction in SMPS particle size for close plumes, going from 113 nm (0.05<f44<0.1) to 101 nm (0.1<f44<0.15). The same trend was observed for the organic aerosols. Considering both AMS and SMPS data for distant fires, there was no considerable change in diameter with aging from less aged BB plumes  to more

aged BB air masses. The particle modes show only slight differences between different f44 bins. This is not consistent with the observed increase in OA for distant fires. Changes in size distribution will be discussed further in the Sect. 3.2.4 where results related to specific BB events are presented.

### 3.2.4 Biomass burning events

Five single source BB events were analysed here (Fig. 2). These episodes were within previously defined close BB periods.

Mass spectra for selected signatures and their time of detection are given in Fig. S6. The spectral signature is similar for these events, with prominent BB-related signals at m/z 60, 73, 29 and 39. However, a significant variation in m/z fragments, especially in m/z 44, can be observed. More detail about these BB events is given in Table 1 and discussed in the following

paragraphs. Different factors were considered including f44, time of day, $\Delta O_3/\Delta CO$, f60, f60/f73, organic concentration and MCE.

On 30[th] of May at around 2pm three hot spots (two having confidence level of approximately 50% and one of 70%) were detected within 2km on the NE from the ATARS (Fig S7a-Fig.S7d). These hot spots likely illustrated two fire events. On the same day and time, 11km on the SE from the sampling site, cluster of events was observed, including 4 hot spots with the confidence level between 94% and 100% and one of 78% confidence level. As all of them were spotted at the same time and within 1km distance, it is most likely that the one big fire has occured. No other close events were observed over this time period. Cluster of hot spots was detected on the SE approximately 50km from the ATARS and big clusters at 100km and 150km, as well as on the SE. Moreover, 200km on E along the backtrajectories cluster of hot spots was observed.

Two single source events, A and C, were identified and analysed here (Fig.2). The event A illustrated the increased signals in the afternoon hours, when wind was coming from the NE. Therefore, the event A is likely result of the hot spots detected within 2km from the ATARS (Fig S7a and Fig S7b). According to backtrajectories air masses during these time period were passing over the land affected by these fires and they were transported from the fire events to ATARS within approximately 10 min. The average organic concentration was found to be 23 μg m$^{-3}$, with the concentrations going up to 45 μg m$^{-3}$. High average MCE value of 0.97 likely suggests flaming fire as emission source and/or grass as a burning material (Desservettaz et al., 2016). The signal named as a C event was detected over the night (Table 1). At that time wind direction was abruptly changed from the NE to SE. Therefore, the signal can be likely associated to cluster of fires 11 km on the SE (Fig S7c and Fig S7d). According to the backtrajectories air masses reached the ATARS within 20 min from the fire cluster. Comparing to the event A, the average organic concentration for the event C was more than threefold with the maximum value of approximately 130 μg m$^{-3}$. As MCE was found to be similar (0.98) to event A value, most likely higher signals illustrate larger fire. The number of hot spots detected on the same time with high confidence level confirms this. Higher fire intensity can also be a reason for the higher f60 values for the event C (0.027) compared to event A (0.016) (Table 1).

On 25[th] of June three hotspot clusters were observed close to ATARS (2km on E, 5km on NE and 10km on SE) (Fig S7e-Fig S7h). The cluster of hot spots observed 10 km from the sampling site had one of the highest hot spot's power (energy released by the fire) observed close to the ATARS (within 20 km) during the campaign (120 MW/km$^2$). Besides the close fires two big clusters around 60 km and 120 km on the SE from ATARS were detected on the same day. On the 25[th] of June, two single-source events were observed and labelled as events F and G. Event F was related to a close fire (two hot spots of 60 and 100% confidence level, within 2 km on E) whose emissions were clearly visible from ATARS (Fig. S7e and Fig S7f). The detected plumes included a considerable portion of oxygenated OA (average f44 value of 0.13, up to 0.18) that could be caused by high daytime photochemical activity. The BB event F began at noon and ended at approximately 5 pm, i.e., it occurred during the period of highest photochemical activity. The relatively high $\Delta O_3/\Delta CO$ ratio (0.13) was elevated relative

to the other close BB events, which suggests additional photochemical activity in the plume. This value falls into the range for fire emissions aged less than 1-2 days in tropical/subtropical regions (Jaffe and Wigder, 2012). However, the f44 value was highly variable during the event F, ranging from 0.07 to 0.18. Other parameters including f60 and organics varied as well. On that time, the wind direction significantly varied between 140° and 80° and likely influenced changes in detected air mass. One of the explanations can be detection of fresh plume and aged masses coming from the distant fires with the change in wind. The low correlation between f60 and f73 might also be an indicator for detection of different BB air masses (Fig. S8). Relatively low organic mass loading and f60 values may be due to highly variable wind direction and/or area rather than combustion conditions (MCE of 0.93), as higher organics are expected for this MCE value. In contrast, event G resulted in the highest organic mass loading measured at ATARS (PM$_1$ concentration of ~350 μg m$^{-3}$). Moreover, the highest chloride concentration observed over the campaign was at the time of G event. The source of emissions was probably within 10 km on SE of the sampling site (Fig S7g and Fig S7h) and the fire magnitude was illustrated by a large burned area observed the following morning (Mallet et al., 2016) and high power of fire observed (30 MW/km$^2$). Backtrajectories analysis indicated that less than half an hour was needed for the air mass to reach the ATARS. High fire intensity was accompanied with smouldering burning conditions (MCE of 0.90) that resulted in high organic loadings and high levoglucosan concentrations (high f60) (Table 1). A low OOA fraction, f44, $\Delta O_3/\Delta CO$ value (0.01) indicated that the plume was likely not aged, which is to be expected due to the proximity and time of the event (started at 10 pm and finished at 4 am).

Intense signals, including high organic concentrations, were also detected on the 9[th] of June (Fig S7i and Fig S7j). On that day cluster of hot spots (all with the confidence level higher than 70%) was detected 5 km from ATARS. Number of distant hot spots was detected between 100 km and 200 km, on the SE from the ATARS. Mallet et al. (2016) suggested that possible sources of this event E might be close fires, distant fires or a combination of both. According to backtrajectories less than 10 min was enough for air masses to cross the distance between 5 km cluster and the research station. Values for f44 (0.08) and $\Delta O_3/\Delta CO$ (0.02) suggested detection of fresh emissions. Moreover, the high f60 value of 0.032 supported this, as distant fires are not likely to be characterized with such a high portion of levoglucosan-like species. However, the possibility of contribution from distant BB plumes cannot be excluded, especially when considering the particle size distribution during event E. The particle size distribution had a mode of 146 nm, while events F and G showed smaller size distributions with modes of 98 and 88 nm, respectively (Table 1). However, different burning material and conditions could also contribute to the larger size distribution mode during event E.

In general, diversity of biomass burning plumes was illustrated through high variability in chemical signature (e.g. large range of f44, from 0.06 to 0.18) for five intense fire events. Events were characterised with different f60/f73 ratios (varying between 1.2 and 2.5), but there was no trend in the relationship of f60/f73 ratio and burning material/conditions (MCE values) (Fig. S8).

**3.3 Positive Matrix Factorization (PMF)**

Initially PMF was performed to assist with analysis of the aging observations rather than to apportion sources since the main source (BB) is already known. This approach was intended to estimate whether the ratios of the different factors, e.g., fresh BBOA, aged BBOA or OOA, exhibited relationships with the age of separate BB plumes. The diversity of the plumes (including close fire plumes with spectra presented in Fig. S6), however, made PMF analysis difficult. The thousands of fires that occurred during the SAFIRED campaign contributed to a wide range of OA composition that reflect different burning materials, conditions, mixed fresh and aged emissions and processing in the atmosphere. The ratio of m/z 43 and 44 differed between plumes (Fig. S11) and since the AMS mass spectra were dominated by these masses, the PMF analysis returned factors that corresponded to individual plumes, even when distant fires were separately examined. In addition, high residuals during BB events could not be reduced even with an unreasonably large number of factors (Fig. S9). The PMF diagnostic plots can be found in Sect. S3.

Instead PMF analysis was applied on "background" data in order to determine the contribution of BB emissions and SOA to regional ambient $PM_1$ outside of the time periods dominated by BB events. The background time periods were determined by examining the plot of f44 vs. f43 (Fig. S11). Cut-offs of 0.15 for m/z 43 and 0.4 for f44 were chosen to remove the influence of periods dominated by BB events which show up as individual lines in f43 vs. f44 space.

PMF performed on the background OA showed significantly smaller residuals (Fig. S12) and resulted in three factors, including a biomass burning OA factor (BBOA), oxygenated OA (OOA) and a factor that represented isoprene-derived OA (Fig. 7). A two-factor solution did not extract the biogenic isoprene-derived OA factor while a four-factor solution resulted in splitting of profiles (Fig. S13). The biomass burning factor, with distinct m/z 60 and 73 signals, contributed to 24 % of the background aerosol. BBOA was also characterized by fragments related to fresh hydrocarbon-like organics, e.g., m/z 27, 29, 41, 43, 55, and 57, and by prominent OOA-related m/z 28 and 44 signals. The large f44 value (an average f44 of 0.09) (Fig. S4) may indicate either the presence of processed BB aerosol and/or fresh emissions that contain oxygenated OA species possibly due to the burning conditions (Heringa et al., 2012;Weimer et al., 2008). A good correlation was observed between the BBOA factor and Org 60 and CO signals (Fig. 7d and Fig. 7e). An OOA profile with prominent m/z 28 and 44, similar to LV-OOA profiles observed in previous studies, was extracted. Its diurnal trend is marked by a broad daily peak which correlates with the maximum temperature and reflects intense daytime photochemical activity. The OOA dominance (contribution of 47 %) and high degree of oxygenation (an average f44 of 0.23) (Fig. S4) illustrate the significance of OOA for this area. The third PMF factor can be related to background biogenic SOA, and is discussed in more detail in the Sect. 3.3.1.

### 3.3.1 Isoprene-derived OA

Isoprene epoxydiols (IEPOX) are important gas-phase precursors for IEPOX-SOA and are products of isoprene oxidation, mostly in low-$NO_X$ environments (Paulot et al., 2009). Recent studies have showed that PMF performed on AMS OA data can be used to determine total IEPOX contribution to SOA (Robinson et al., 2011;Lin et al., 2011;Budisulistiorini et al., 2013;Hu et al., 2015;Xu et al., 2016). A similar distinct PMF-resolved factor has been extracted in this study. In order to consider other possible pathways in isoprene-derived OA formation (Schwantes et al., 2015), the PMF factor has been named as isoprene-derived OA (Xu et al., 2016;Pye et al.)The isoprene-derived OA mass spectrum can be clearly distinguished from BBOA and OOA by an enhanced m/z 82 signal (Fig. 9a). In the HR analysis of this dataset, $C_5H_6O^+$ is the most dominant peak that contributes to m/z 82 (Fig. S13) and has been confirmed to be a reliable tracer for isoprene-derived OA (Hu et al., 2015). There was a strong correlation observed between the isoprene- derived OA factor and Org 82 ($C_5H_6O^+$) (R of 0.90) (Fig. 7f). The isoprene- derived OA factor profile was also characterized with enhanced peaks at m/z 53, 43 and 44. An AMS spectrum with the same prominent peaks was reported by Allen et al. (2008) for the Darwin region in the Northern Territory. This was observed during the monsoon-break period during the wet season in February, when fires were not common, and clean air of biogenic origin was suggested as a source. Moreover, our spectrum is similar to isoprene-related OA spectra reported previously (Robinson et al., 2011;Lin et al., 2011;Budisulistiorini et al., 2013). The average isoprene concentration measured during the campaign was $0.49 \pm 0.78$ ppb. However, it should be emphasised that monoterpene-derived OA can also influence the background level of $C_5H_6O^+$ (Hu et al., 2015). The average monoterpenes concentration for this study was found to be $0.22 \pm 0.41$ ppb.

The main path for isoprene-derived OA formation is proposed to be acid-catalysed IEPOX uptake (Lin et al., 2011;Lambe et al., 2015). According to calculations for the composition depended collection efficiency, 22% of the aerosol were acidic. In order to estimate whether the acidity of the particles had an influence on isoprene- derived OA generation in ATARS, the correlation between sulfate (taken as proxy of aerosol acidity) and the isoprene-derived OA factor was examined. The correlation between the factor and sulfate can be considered as weak (R of 0.3) (Fig S18). However, two periods can be clearly distinguished from the graph: the period before 5[th] of June and the period after 15[th] of June. While there is no correlation between sulfate and the isoprene-derived OA factor for the first period, when plotting only data collected from 15[th] of June, correlation is found to be slightly higher than the correlation for all background data (R of 0.4).

The isoprene/furan concentrations at m/z 69 (PTR-MS) were treated as an isoprene contribution due to the dominance of isoprene signal compared to furan, according to the samples analysed by gas chromatography-mass spectrometry (Fig. S17). The furan contribution was more significant in BB plumes during the close BB period, as suggested previously (Warneke et al., 2011). As expected, the isoprene/furan gas-phase concentration measured by the PTR-MS increased from noon till the late afternoon (Fig. S16). According to diurnal patterns of PMF isoprene- derived OA, there is no considerable change in isoprene- derived OA concentration from noon throughout the day. The isoprene- derived OA fraction is expected to

increase during daytime hours in accordance with enhancements in isoprene and gas-phase IEPOX (Hu et al., 2015). However, the isoprene- derived OA factor was prominent during night and morning hours. Significant isoprene- derived OA factor enhancement during the night time might be due to partitioning of lower volatility species onto the particles when the temperature drops and relative humidity increases as suggested by Budisulistiorini et al. (2013), due to transport of distant air masses, or lower boundary layer height. During the night, the boundary level lowers which increases the concentration of gaseous compounds and can induce partitioning of gases onto the particles. Therefore, the lower night-time boundary layer might create conditions for low volatility isoprene-derived OA partitioning and an increase in isoprene-derived OA.

A plot of f44 ($CO_2^+$) vs. f82 ($C_5H_6O^+$) (Org $CO_2^+$/Org and Org $C_5H_6O^+$/Org from HR data analysis, respectively) (Fig. 8) introduced by Hu et al. (2015) can be used to distinguish IEPOX-SOA from SOA originating from other sources, including isoprene- derived SOA that are generated from species other than IEPOX. This plot also indicates the degree of aging of the IEPOX-SOA. The general pattern of f44 increase with f82 decrease is observed. With aging, OA becomes more oxidized and the $C_5H_6O^+$ signature decreases. This can be due to oxidation processes or mixing with more aged aerosol masses.

A background value for f82 ($C_5H_6O^+$) in environments strongly influenced by BB was suggested to be 0.0017 and is indicated by the vertical dashed black line (Hu et al., 2015). Looking at our data plotted in f44 vs. f82 a similar observation can be made. All data points are positioned above 0.0016 of f82 with an average value of 0.0061 ± 0.0036. The isoprene-derived OA factor average values for this campaign and two other ambient studies are marked in the plot. The f82 value for isoprene-derived OA factor observed here is similar to the reported f82 value for an urban site (Budisulistiorini et al., 2013), but lower compared to the factor observed for Borneo forest (Robinson et al., 2011). The lower value for SAFIRED compared to Borneo forest, considering that biogenic influence is significant for both environments, can be attributed to the high influence of BB emissions.

As the most abundant NMOC, isoprene is a significant contributor to the global SOA budget (Guenther et al., 2012;Hallquist et al., 2009;Paulot et al., 2009). isoprene-derived OA during the SAFIRED campaign accounted for 28 % of the total background OA which is similar to previous observations where the isoprene-derived OA fraction contribution varied from 6-36 %, depending on the environment (Hu et al., 2015). This confirms the importance of isoprene-derived SOA for the Northern Territory environment even in times of high BB influence.

## 4. Conclusions

A one month long campaign called SAFIRED was conducted in northern Australia during a period of significant burning (early dry season). There was a significant influence of BB plumes on the atmospheric chemical profile at the time of campaign, reflected by high concentrations of gaseous and particle species including CO and $PM_1$ organics, reaching maxima of ~$10^4$ ppb and 350 µg m$^{-3}$ respectively during heavy BB episodes. Emitted aerosols were predominantly organic

species (90 %) with a wide range of levels of oxidation. There was a clear overall increase of the highly oxygenated OA fraction and degradation of BB-related signatures with OA processing. This was shown by an overall trend of f44 increase and f60 decrease for distant fires. Plume emissions formed over the month period were photochemically active resulting in the production of tropospheric ozone. Close fire emissions were characterized by lower ozone enrichments (average

$\Delta O_3/\Delta CO$ of 0.15) than plumes detected from the distant BB sources (average $\Delta O_3/\Delta CO$ of 0.31) which illustrate higher photochemical activity with more processed air masses. This emphasizes air mass ability for photochemical processing and production of SOA. An increase in the $\Delta OA/\Delta CO$ ratio with increase of f44 also suggests SOA formation. According to results, the OA oxidation level did not significantly influence particle size distribution. Diversity in BB emissions was illustrated through investigation of five selected events. The chemical signature varied for different fire events (e.g. wide

range in f44, from 0.06 to 0.18).

PMF was employed to estimate BB influence on and SOA portion of background regional aerosol. A significant portion of oxygenated OA, identified through high f44 (majority of data points between 0.1 and 0.25), and a significant portion of aged BBOA (47 and 24 %, respectively) were observed. The latter suggests considerable processing of aerosol in the atmosphere for this area. The remaining OA was attributed to isoprene-derived OA factor (28 %), identified here for the first time in

Australia. The OA mass spectrum with isoprene-derived OA characteristics previously reported in the wet season and now observed with prominent BB influence during the dry period, suggests the importance of biogenic isoprene sources for the Northern Territory area at all times of the year.

Observed photochemical activity of air masses and enhancement of OA with aging illustrate the importance of aging and SOA formation in the Northern Territory during the dry early season. SOA is recognized as a significant contributor to

climate, environment and adverse human health effects. This study is an important step in addressing suggested further research related to tropical biomass burns and biogenic-related SOA in Australia (Rotstayn et al., 2009). As SOA yields are considerably underestimated and its formation is still not sufficiently understood, this study can facilitate understanding of SOA formation for Northern Territory savannah areas and also for the tropics in general. Additional measurements during late dry season (September-October) are needed as more intense and frequent fires occur during this period (Andersen et al.,

2005;Williams et al., 1998). Moreover, the late dry season is suggested to have more aged emissions (Ristovski et al., 2010;Wardoyo et al., 2007). Therefore it is important to characterize SOA formation, and yield during this period.

**Data availability.** The underlying research data can be accessed upon request to the corresponding author (Branka Miljevic; b.miljevic@qut.edu.au).

**Author contributions.** Andelija Milic analysed and interpreted the data and prepared the manuscript. Marc Mallet operated

the cToF-AMS and contributed to data analysis, interpretation and writing. Branka Miljevic installed the cToF-AMS, contributed to data analysis, interpretation and writing and supervised the work of Andelija Milic. Luke Cravigan set up the

SMPS and contributed to data interpretation. Joel Alroe assisted in organizing the QUT instrumentation, data interpretation and writing the manuscript. Zoran Ristovski contributed to campaign organization and data interpretation and supervised the work of Andelija Milic. Leah Williams contributed to setting up the cToF-AMS, preliminary data analysis and data interpretation. Melita Keywood organized and led the campaign. Paul Selleck operated the BAM, analysed the data and contributed to data interpretation. Sarah Lawson operated the PTR-MS and analysed the data. James Ward operated the Ozone analyser and analysed the data. Maximilian Desservettaz operated the FTIR and SMPS and analysed the data. Clare Paton-Walsh contributed to campaign organization and running the campaign. All authors declare that they have no conflict of interest.

**Acknowledgements.** The authors thank Manjula Canagaratna for valuable discussions about interpreting the PMF analysis. The authors thank Min Cheng for providing the gas chromatography-mass spectrometry data. This work was supported by the Australian Research Council Discovery grant (DP120100126).

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

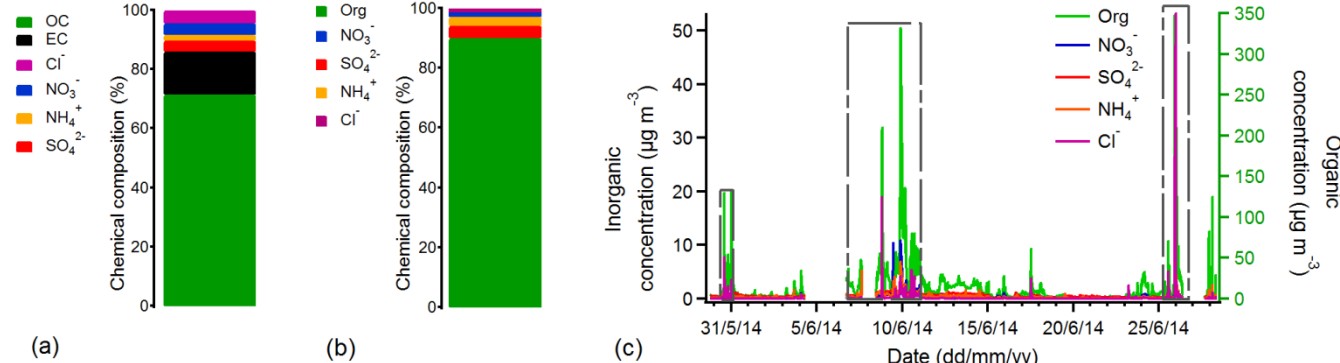

**Figure 1: Contributions of (a) EC, OC and inorganics collected on filters and the contribution of (b) AMS organic and inorganics in non-refractory submicron aerosol fraction as well as (c) their time series. Close BB periods are marked in the AMS time series with grey dashed boxes. The distant BB periods cover all days of the measurement other than days included in close BB periods. Gaps in the AMS time series indicate gaps in the sampling.**

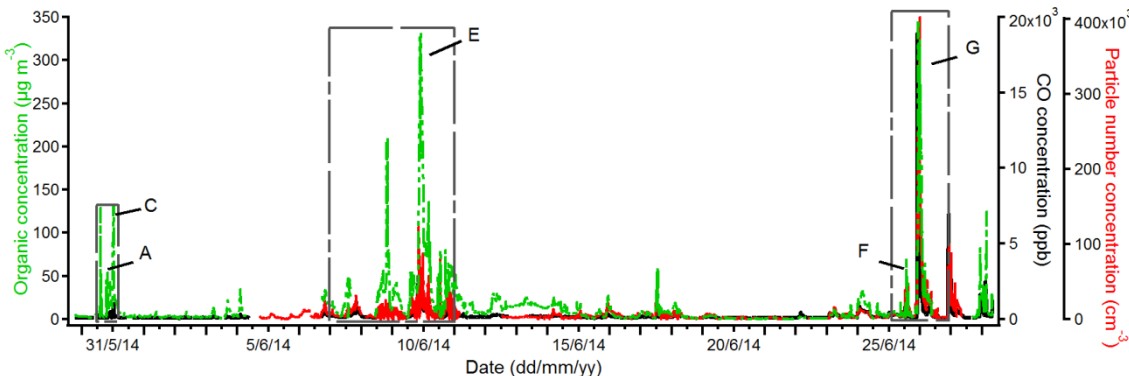

**Figure 2: Time series for AMS organics, CO and particle number concentration during the SAFIRED campaign. Close BB periods are marked with grey dashed boxes and selected BB events are labelled. The distant BB periods cover all days of the measurement other than days included in close BB periods. Gaps in the time series indicate gaps in the sampling (no data for the instrument).**

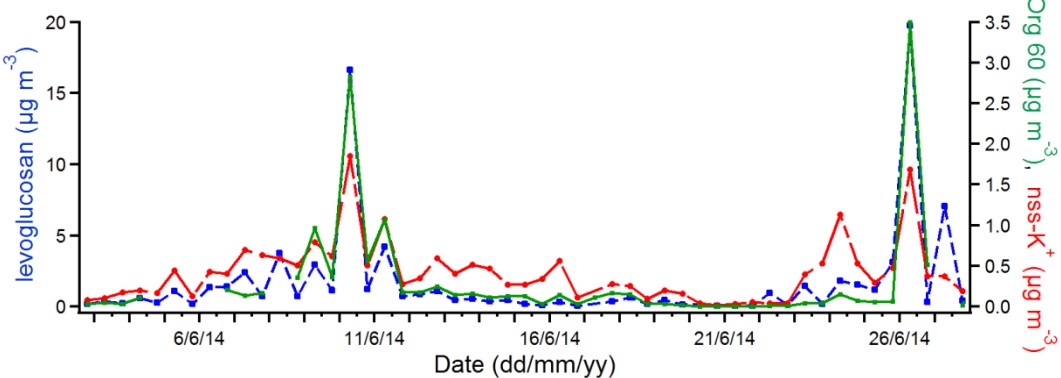

**Figure 3: Time series of BB tracers: levoglucosan, soluble non-sea salt potassium (nss-K⁺) (both 12h resolution) and AMS Org 60 (averaged to BAM 12h).**

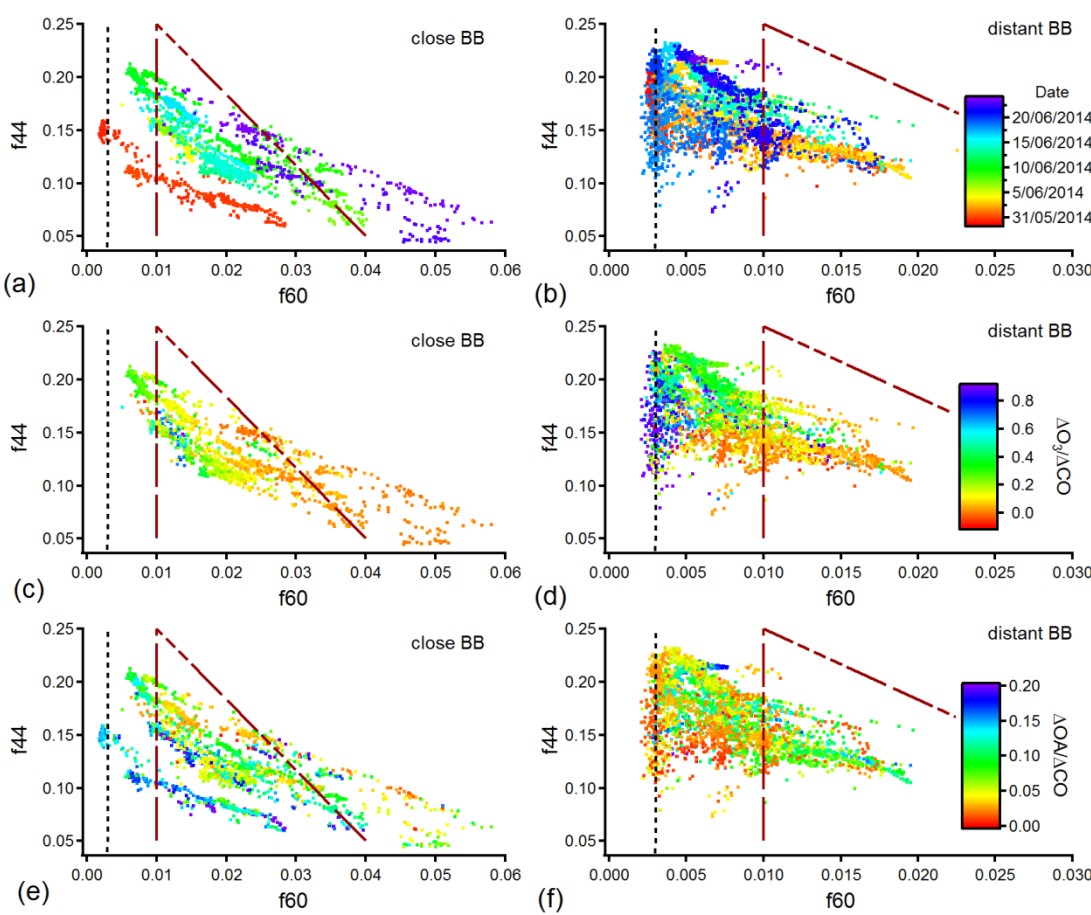

Figure 4: f44 vs. f60 coloured by date for (a) close and (b) distant BB periods, by $\Delta O_3/\Delta CO$ ratio for (c) close and (d) distant BB periods, and by $\Delta OA/\Delta CO$ ratio for (e) close and (f) distant BB periods (vertical black lines refer to the f60 background level of 0.003). Red dashed lines refer to the ambient BBOA-related data introduced by (Cubison et al., 2011). Note: ozone data from 29th of May until the 1st of June were not available.

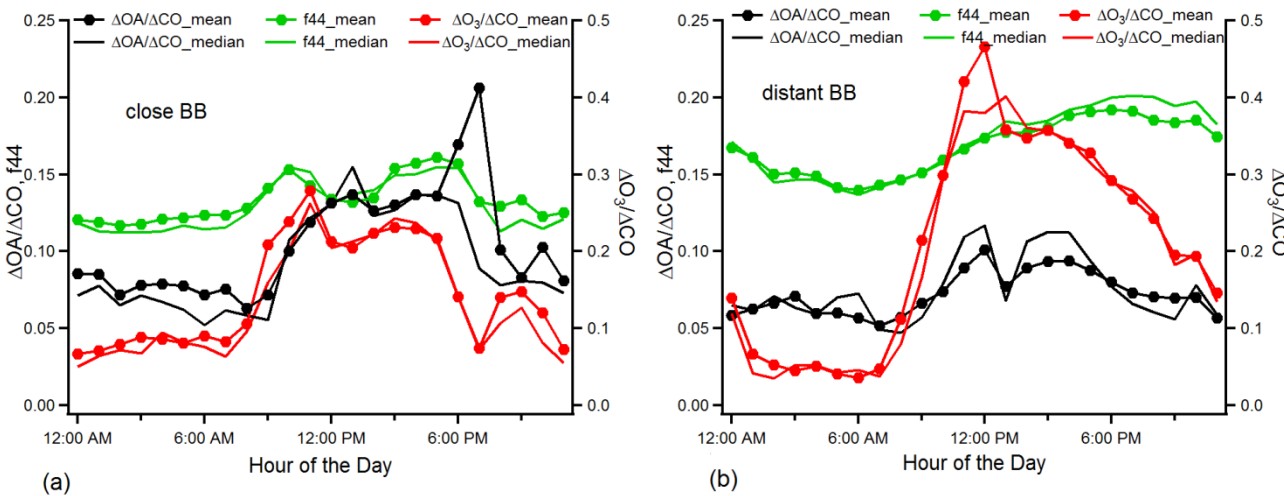

**Figure 5: Diurnal trend of f44 (mean and median), ΔO₃/ΔCO ratio (mean and median) and ΔOA/ΔCO ratio (mean, median)for close (a) and distant (b) BB periods.**

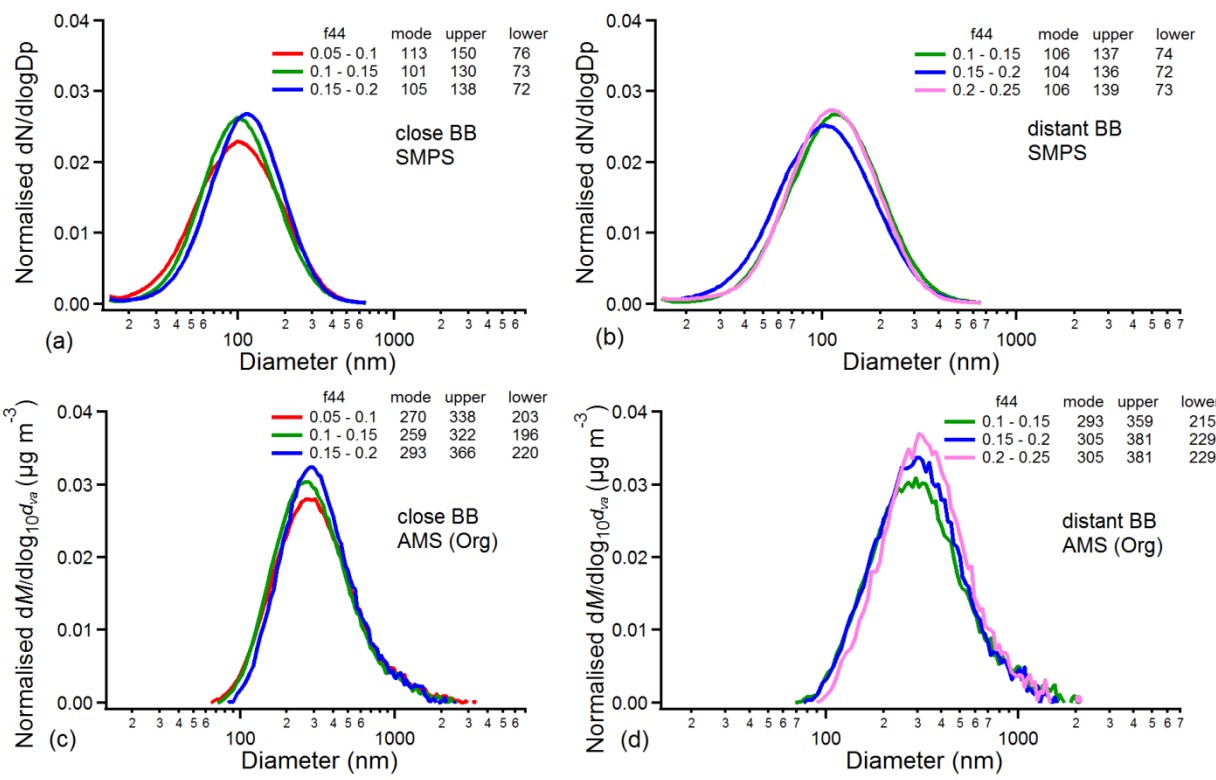

**Figure 6: SMPS size distributions (dN/dlogDp, normalised to total particle number concentration, versus diameter) binned by the AMS f44 for close (a) and distant (b) BB periods and AMS size distributions (dM/dlog$_{10}$d$_{va}$, normalised to total organic concentration, versus diameter) binned by the AMS f44 for close (c) and distant (d) BB periods.**

5    **Table 1: Selected BB event values for f44, f60, organic concentration, CO, MCE, mode diameter and ΔO$_3$/ΔCO ratio, along with measurement start and end time. ND refers to no data.**

| | Date (start/end) | f44±SD (range) | mean f60±SD (range) | mean Org±SD* (range) | mean CO±SD* (range) | MCE | ΔO$_3$/ΔCO | Mode diameter (nm) |
|---|---|---|---|---|---|---|---|---|
| A | 30/05/14 18:34- 30/05/14 19:25 | 0.087±0.08 (0.079-0.105) | 0.016±0.002 (0.010-0.018) | 22.6±12.5 (4.6-45.2) | 185.9±95.5 (90.2-370.1) | 0.97± 0.06 | ND | ND |
| C | 30/05/14 23:41- 31/05/14/ 00:59 | 0.066±0.005 (0.060-0.079) | 0.027±0.002 (0.021-0.028) | 75.5±40.0 (13.7-131.2) | 627.1±345.4 (185.8-1181.6) | 0.98± 0.11 | ND | ND |
| E | 09/06/14 19:45- 10/06/14 00:32 | 0.078±0.013 (0.062-0.093) | 0.032±0.002 (0.030-0.035) | 175.9±105.0 (87.3-331.8) | 1558.5±965.5 (671.6-3382.5) | 0.91± 0.05 | 0.024 | 146 |
| F | 25/06/14 12:28- 25/06/14 16:59 | 0.134±0.031 (0.073-0.178) | 0.009±0.002 (0.007-0.014) | 13.2±14.5 (2.4-70.8) | 479.0±348.8 (139.5-1642.7 | 0.93± 0.04 | 0.134 | 98 |
| G | 25/06/14 21:40- 26/06/14 03:59 | 0.062±0.017 (0.045-0.098) | 0.046±0.004 (0.035-0.052) | 144.6±104.7 (25.8-347.7) | 2744.6±2299.8 (592.8-11275.7) | 0.90± 0.06 | 0.011 | 88 |

*unit is μg m$^{-3}$

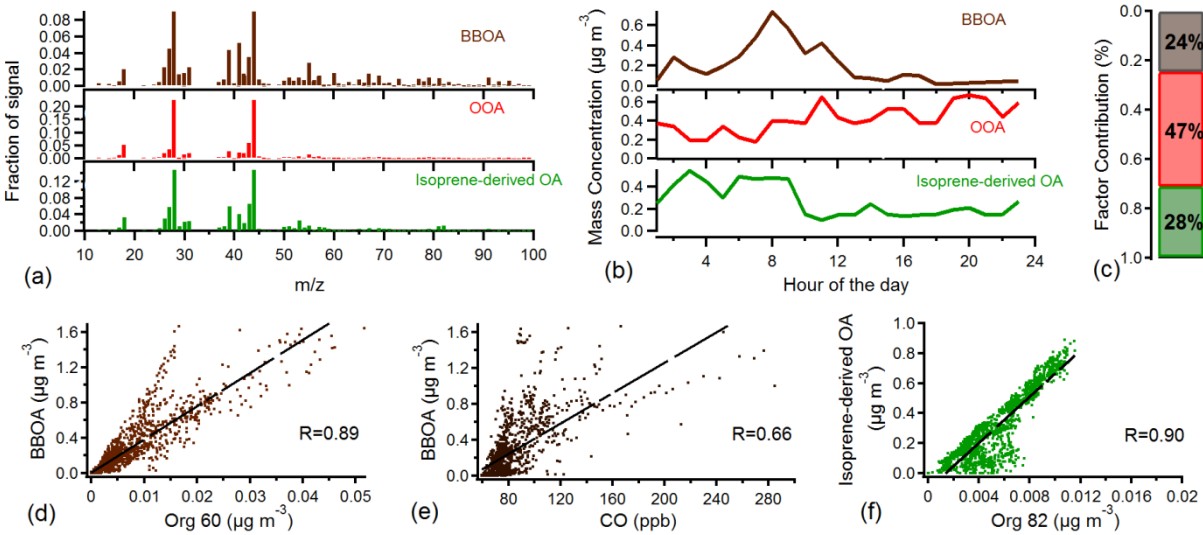

**Figure 7:** (a) Mass spectra, (b) diurnal trends, (c) contribution and (d), (e), (f) correlations for the PMF 3-factor solution for background periods including BBOA, OOA and isoprene-derived OA factors.

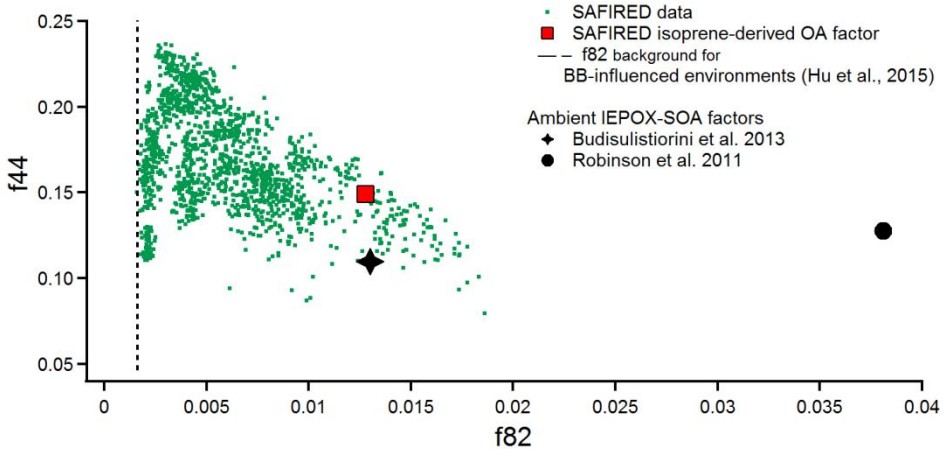

**Figure 8:** f44 ($CO_2^+$) vs. f82 ($C_5H_6O^+$) for the SAFIRED data; The isoprene-derived OA factor from SAFIRED and two other ambient campaigns (Budisulistiorini et al., 2013;Robinson et al., 2011) are marked. Vertical black dashed line indicates proposed background f82 value in environments strongly influenced by BB (Hu et al., 2015).