# Peer review of "Fresh and aged aerosols emitted from biomass burning in northern Australia"

_Atmospheric Chemistry and Physics, 2016_

## Referee Comment (RC1) · Anonymous Referee #1 · 14 Sep 2016

Review of "Aging of aerosols emitted from biomass burning in northern Australia" by Milic et al.

This manuscript studied the biomass burning organic aerosol in Australia by using a c-ToF-AMS. The heavy biomass burning causes that 90% of NR-PM$_1$ is organics. Five intense BB events with different burning conditions are analyzed. The enhancement of ozone and SOA in biomass burning plumes in examined. PMF analysis on background OA resolved three factors, BBOA, OOA, and IEPOX-SOA. The IEPOX-SOA is identified for the first time in Australia. The manuscript is well structured and the content is appropriate for ACP. However, most of the work presented here has been done before. The manuscript in its current form does not add significantly to the literature. Thus, I recommend major revisions. More in depth analysis and conclusions are required.

Major comments.

1.      Whether this study is suitable/capable to study the aging of biomass burning OA.

Previous field studies on this topic usually used aircraft to follow the biomass burning plumes, which ensures investigating the same air mass (Cubison et al., 2011; Yokelson et al., 2009; Forrister et al., 2015; May et al., 2015). However, this is not the case in this study as the air mass is changing. In fact, many observations in this study can be explained by changes in air mass or physical changes, instead of aging of biomass burning OA. For example, an increase in $f$44 and a decrease in $f$60 (Figure 4b) could be simply due to an BBOA-rich air mass mixing with OOA-rich background air or physical changes during dilution (May et al., 2015). This could be potentially revealed in Figure 4a (close BB) if the authors color the data points by OA concentration. The data points with higher $f$60 and lower $f$44 would have higher OA concentration.

The authors may use MODIS sensor and backtrajectory analysis (section 2.1.7) to pinpoint fire position and calculate transport time. Then, the oxidation rate of BBOA or its tracers (such as $f$60) can be calculated, which is highly uncertain now. The aging time may also be estimated by using the photochemical clock, such as NO$_x$/NO$_y$ ratio.

The OA in the five biomass burning events (section 3.2.4) is dominantly from fresh biomass burning, which provides a good opportunity to study the properties of fresh BBOA. The study would benefit from expanding related discussions. For example, "different burning

conditions" are vaguely mentioned multiple times to explain the difference between the five biomass burning events. However, can the authors be more explicit about the relationship between specific burning conditions/burning materials with OA concentration, $f44$, $f60$, $\Delta O_3$, etc? It is helpful to bring Table S2 to the main text.

2.    IEPOX-SOA related.

Firstly, it is not appropriate to label the PMF factor as IEPOX-OA. While the majority of this factor is from the reactive uptake of IEPOX, this factor also likely includes contribution from isoprene OA formed via other pathways. For example, a recent study by Schwantes et al. (2015) showed clearly that SOA formed via the reactive uptake of isoprene nitrooxy hydroxyepoxide could also produce fragment $m/z$ 82. In addition, the mass concentration of this PMF factor cannot be fully explained by the total IEPOX SOA tracers (Hu et al., 2015; Budisulistiorini et al., 2015). Thus, it is more appropriate the name the factor as isoprene-derived OA (Xu et al., 2016; Pye et al., 2016).

Secondly, since the isoprene-OA factor is identified for the first time in Australia, more analysis could be done related to this factor. For example, does this factor correlate with sulfate, which has been shown in previous studies (Xu et al., 2015a; Budisulistiorini et al., 2015)? What's the NO concentration at the measurement site? Is IEPOX formed locally or from transport? What's the particle pH? As shown in Figure 9f, in general, IEPOX-OA correlates well with Org 82. However, some data are scattered (i.e., Org 82 0.004 to 0.008 µg m$^{-3}$ range). What happens for these data? Is it due to sources other than IEPOX uptake contributing to Org 82?

Minor comments.

1.    Page 2 Line 6. Elemental carbon (EC) is not the same as black carbon (BC). EC is defined by thermal properties and BC is defined by optical properties (Andreae and Gelencsér, 2006).

2.    Page 4 Line 4. It is stated that sampled air mass had mainly passed over land. However, this seems to contradict with the conclusion in Page 8 Line 18-20, that a significant portion of chlorides could be sea salt.

3.    Page 4 Line 19-20. It is not clear if the CE is determined by using the UMR data or the HR data. Or HR data are used in the squirrel panel? Also, sulfate fragments should not have

interference from organic fragments in HR data. Thus, HR data should be used throughout the analysis.

4. Page 7 Line 31. It is not clear how the "close BB" and "distant BB" periods are separated. The criteria for separation should be clearly stated?

5. Page 8 Line 14. The authors can use the $NO^+/NO_2^+$ ratio to infer if the nitrate functionality originates from inorganic or organic nitrate (Xu et al., 2015b; Farmer et al., 2010).

6. Page 10 Line 22-23. This sentence is really confusing and should be re-written.

7. Page 11 Line 22. Is the OA mass loading related to wind speed or wind direction?

8. Page 12 Line 14. The authors may check if the relationship between $f$60 and $f$73 changes with burning conditions or burning materials.

9. Page 12 Line 31. Is there any study to show that fresh emissions from biomass burning contain OOA?

10. Page 13 Line 10. Cite Xu et al. (2015a).

11. Page 14 Line 8-11. The low correlation between IEPOX-OA and BBOA could be simply due to that these two factors are from completely different sources and a correlation is not expected. The discussion in this sentence is not supported and should be removed.

Reference

Andreae, M. O., and Gelencsér, A.: Black carbon or brown carbon? The nature of light-absorbing carbonaceous aerosols, Atmos. Chem. Phys., 6, 3131-3148, 10.5194/acp-6-3131-2006, 2006.

Budisulistiorini, S. H., Li, X., Bairai, S. T., Renfro, J., Liu, Y., Liu, Y. J., McKinney, K. A., Martin, S. T., McNeill, V. F., Pye, H. O. T., Nenes, A., Neff, M. E., Stone, E. A., Mueller, S., Knote, C., Shaw, S. L., Zhang, Z., Gold, A., and Surratt, J. D.: Examining the effects of anthropogenic emissions on isoprene-derived secondary organic aerosol formation during the 2013 Southern Oxidant and Aerosol Study (SOAS) at the Look Rock, Tennessee ground site, Atmos. Chem. Phys., 15, 8871-8888, 10.5194/acp-15-8871-2015, 2015.

Cubison, M. J., Ortega, A. M., Hayes, P. L., Farmer, D. K., Day, D., Lechner, M. J., Brune, W. H., Apel, E., Diskin, G. S., Fisher, J. A., Fuelberg, H. E., Hecobian, A., Knapp, D. J., Mikoviny, T., Riemer, D., Sachse, G. W., Sessions, W., Weber, R. J., Weinheimer, A. J., Wisthaler, A., and Jimenez, J. L.: Effects of aging on organic aerosol from open biomass burning smoke in aircraft and laboratory studies, Atmos. Chem. Phys., 11, 12049-12064, 10.5194/acp-11-12049-2011, 2011.

Farmer, D. K., Matsunaga, A., Docherty, K. S., Surratt, J. D., Seinfeld, J. H., Ziemann, P. J., and Jimenez, J. L.: Response of an aerosol mass spectrometer to organonitrates and organosulfates and implications for atmospheric chemistry, Proceedings of the National Academy of Sciences, 107, 6670-6675, 10.1073/pnas.0912340107, 2010.

Forrister, H., Liu, J., Scheuer, E., Dibb, J., Ziemba, L., Thornhill, K. L., Anderson, B., Diskin, G., Perring, A. E., Schwarz, J. P., Campuzano-Jost, P., Day, D. A., Palm, B. B., Jimenez, J. L., Nenes, A., and Weber, R. J.: Evolution of brown carbon in wildfire plumes, Geophysical Research Letters, n/a-n/a, 10.1002/2015GL063897, 2015.

Hu, W. W., Campuzano-Jost, P., Palm, B. B., Day, D. A., Ortega, A. M., Hayes, P. L., Krechmer, J. E., Chen, Q., Kuwata, M., Liu, Y. J., de Sá, S. S., McKinney, K., Martin, S. T., Hu, M., Budisulistiorini, S. H., Riva, M., Surratt, J. D., St. Clair, J. M., Isaacman-Van Wertz, G., Yee, L. D., Goldstein, A. H., Carbone, S., Brito, J., Artaxo, P., de Gouw, J. A., Koss, A., Wisthaler, A., Mikoviny, T., Karl, T., Kaser, L., Jud, W., Hansel, A., Docherty, K. S., Alexander, M. L., Robinson, N. H., Coe, H., Allan, J. D., Canagaratna, M. R., Paulot, F., and Jimenez, J. L.: Characterization of a real-time tracer for isoprene epoxydiols-derived secondary organic aerosol (IEPOX-SOA) from aerosol mass spectrometer measurements, Atmos. Chem. Phys., 15, 11807-11833, 10.5194/acp-15-11807-2015, 2015.

May, A. A., Lee, T., McMeeking, G. R., Akagi, S., Sullivan, A. P., Urbanski, S., Yokelson, R. J., and Kreidenweis, S. M.: Observations and analysis of organic aerosol evolution in some prescribed fire smoke plumes, Atmos. Chem. Phys., 15, 6323-6335, 10.5194/acp-15-6323-2015, 2015.

Pye, H. O. T., Murphy, B. N., Xu, L., Ng, N. L., Carlton, A. G., Guo, H., Weber, R., Vasilakos, P., Appel, K. W., Budisulistiorini, S. H., Surratt, J. D., Nenes, A., Hu, W., Jimenez, J. L., Isaacman-VanWertz, G., Misztal, P. K., and Goldstein, A. H.: On the implications of aerosol liquid water and phase separation for organic aerosol mass, Atmos. Chem. Phys. Discuss., 2016, 1-44, 10.5194/acp-2016-719, 2016.

Schwantes, R. H., Teng, A. P., Nguyen, T. B., Coggon, M. M., Crounse, J. D., St. Clair, J. M., Zhang, X., Schilling, K. A., Seinfeld, J. H., and Wennberg, P. O.: Isoprene NO3 Oxidation Products from the RO2 + HO2 Pathway, The Journal of Physical Chemistry A, 10.1021/acs.jpca.5b06355, 2015.

Xu, L., Guo, H., Boyd, C. M., Klein, M., Bougiatioti, A., Cerully, K. M., Hite, J. R., Isaacman-VanWertz, G., Kreisberg, N. M., Knote, C., Olson, K., Koss, A., Goldstein, A. H., Hering, S. V., de Gouw, J., Baumann, K., Lee, S.-H., Nenes, A., Weber, R. J., and Ng, N. L.: Effects of anthropogenic emissions on aerosol formation from isoprene and monoterpenes in the southeastern United States, Proceedings of the National Academy of Sciences, 112, 37-42, 10.1073/pnas.1417609112, 2015a.

Xu, L., Suresh, S., Guo, H., Weber, R. J., and Ng, N. L.: Aerosol characterization over the southeastern United States using high-resolution aerosol mass spectrometry: spatial and seasonal variation of aerosol composition and sources with a focus on organic nitrates, Atmos. Chem. Phys., 15, 7307-7336, 10.5194/acp-15-7307-2015, 2015b.

Xu, L., Middlebrook, A. M., Liao, J., de Gouw, J. A., Guo, H., Weber, R. J., Nenes, A., Lopez-Hilfiker, F. D., Lee, B. H., Thornton, J. A., Brock, C. A., Neuman, J. A., Nowak, J. B., Pollack, I. B., Welti, A., Graus, M., Warneke, C., and Ng, N. L.: Enhanced formation of Isoprene-derived Organic Aerosol in Sulfur-rich Power Plant Plumes during Southeast Nexus (SENEX), Journal of Geophysical Research: Atmospheres, 2016JD025156, 10.1002/2016JD025156, 2016.

Yokelson, R. J., Crounse, J. D., DeCarlo, P. F., Karl, T., Urbanski, S., Atlas, E., Campos, T., Shinozuka, Y., Kapustin, V., Clarke, A. D., Weinheimer, A., Knapp, D. J., Montzka, D. D., Holloway, J., Weibring, P., Flocke, F., Zheng, W., Toohey, D., Wennberg, P. O., Wiedinmyer, C., Mauldin, L., Fried, A., Richter, D., Walega, J., Jimenez, J. L., Adachi, K., Buseck, P. R., Hall, S. R., and Shetter, R.: Emissions from biomass burning in the Yucatan, Atmos. Chem. Phys., 9, 5785-5812, 10.5194/acp-9-5785-2009, 2009.

---

## Referee Comment (RC2) · Anonymous Referee #2 · 18 Sep 2016

This paper discussed the aging of biomass burning (BB) aerosols from Northern territory of Australia based on aerosol mass spectrometer (AMS) at a ground observation site. The entire field study (29 May to 20 June, 2014) was divided to be "close BB" and "distant BB" periods. The BB aerosols were investigated by comparing aerosol chemical composition, aging process, ozone formation, size distribution and SOA formation between those two periods. PMF analysis was performed on the background OA period. Ambient Isoprene Epoxydiols-Derived SOA (IEPOX-SOA) factor from three PMF-factor solution was discussed and showed similar performances to the same factor reported in the previous studies. This paper showed a good dataset for investigating the biomass burning influences on aerosols. However, the analysis in this paper is not clear and is not presented in a quite logic way. Thus I recommend a major revision for this paper before considering its publication in ACP.

1. The authors analyzed the data in section 3.2 based on dividing the entire study to be "distant BB" and "close BB" periods in section 3.1. However, it is not clear that how the "close BB" and "distant BB" periods were defined. Was it based on distance or the emission intensity, or the correlation between organic, CO and particle number concentration? The authors only labelled "close BB" period in Figs.1-2, "distant BB" period and background OA period are unknown.

2. Suggest revising the structure of the results and discussion part. After all comparisons between the "close BB" and "distant BB" being discussed in section 3.2.1-3.2.3, the paper started to describe the details about the "close BB" period in section 3.2.4, which is very confusing. I do not quite understand what the relationship between the 5 BB events in section 3.2.4 with the "close BB" period analyzed in section 3.2.1-3.2.3 is. I suggest the authors introduce the details on "close BB" and "distant BB" period first, then discuss the general comparison results. In this way, the readers can get a rough idea on what "close BB" and "distant BB" are then understand later comparison better. I also suggest the authors add a separate figure for readers to see the details of those "close BB" and "distant BB" periods. In this way, the good correlation addressed in page 7 Line 30 can also be understood.

3. Page 8 Line 10-line 20:

1) The authors mentioned there were high chloride (KCl or NH₄Cl) mass concentrations from the biomass burning, then said a significant portion of chlorides from BAM measurement came from sea salt. Those two statements are conflictive.
2) If the signals are strong, $KCl^+$ and $NaCl^+$ ions can be resolved in the OA spectra in the HR fitting of PIKA.
3) The chloride with high concentrations in Fig.1 was not shown in the comparison plot of Fig. S2. Were some periods missed in Fig. S2? The authors should point out which periods were used for comparison in Fig.S2.
4) Line 10, "This can be explained by the depletion of chloride with transport and aging of BB plumes (Li et al., 2003;Li et al., 2010;Liu et al., 2000)." There is no proof for this sentence. Since the author has the K+ and Na+ measurement, the authors can calculate the chloride depletion fraction to show the evidence for this sentence. See equation (A1) and Fig. (A2) in (Hayes et al., 2013) for reference.

4. Page 9 line 7, "The variability observed in f44 vs. f60 for close BB events (Fig. 4a) probably reflect BB plumes generated during different burning conditions rather than different atmospheric processing of BB masses." There is no evidence to support this sentence. In Fig. 5a, higher $\Delta O3/\Delta CO$ was observed for the high f44. The authors explained this figure as (page 9 line 31): "On average, the $\Delta O3/\Delta CO$ ratio increases with f44 and decreases with f60, indicating increased photochemical processing of OA in plumes with atmospheric aging and ozone production". Two statement are contradicting with each other.
5. Page 9 line 14 "This confirms that levoglucosan-like species carried by the BB plume did not degrade to background levels even as oxidized species were formed. Thus, f60 is a reasonable marker of distant BB in this study." This expression is not quite true, lots of point in Fig 4b are around background level when the f44 is very high. Mixing BB plumes with other aged plumes also can degrade the f60.

6. Page 9 line 23 "The wide range in f44 can be attributed to difference in burning conditions for close BB". What the author missing in this entire section is dilution or mixing BB plumes with other plumes. Not only different burning condition or aging, but also the dilution and mixing will also lead the similar evolution trend of f44 vs f60 in the triangle plot (Fig. 4).

7. Page 10 line 2. Suggest the authors use $Ox(=NO_2+O_3)$ instead of $O_3$, or use both, then the $O_3$ loss by NO can be accounted for (Herndon et al., 2008).

8. Page 10 line 6-7, How did the authors calculate the average $\Delta O_3/\Delta CO$ from different fires? How did the authors obtain $\Delta O_3/\Delta CO$ from "distant BB" fire plumes since the plume spikes are not obvious as those in "close BB" period?

9. Page 10 line 14, The authors should calculate $\Delta OA/\Delta CO$ vs $\Delta f44$ (not f44), because it is not clear if high f44 can really reflect the SOA aging since different BB plumes may have different f44 due to burning condition and mixing or dilution. I suggest the authors check the photochemical age calculated from VOCs (de Gouw et al., 2005), which can be another parameter for indicating oxidation/aging process of plumes. And also the authors have a great dataset to see emission ratio of BB emitted aerosols (e.g. $\Delta SO_4/\Delta CO$, $\Delta NH_4/\Delta CO$ and $\Delta Cl/\Delta CO$).

10. Page 10 3.2.3 section. Suggest adding size distributions of m/z 44 and m/z 60, which can be used to compare with the size distributions from SMPS. This comparison can help to interpret the comparison between "low f44" and "high f44" periods. In addition, the comparison of size distributions between "close BB" and "distant BB" can be added. This can help to characterize those two periods and see if the "distant BB" are really more aged or mixed with aged aerosols.

11. Page 14 line 1-2, what is the monoterpene concentration in this study? Because monoterpene-derived SOA also can influence the background level of $C_5H_6O^+$ (Hu et al., 2015). Isoprene concentration was not reported in this study, which is important for discussing IEPOX-SOA.

11. Page 14 8-9. "low correlation between the IEPOX-SOA time series with both OOA and BBOA was observed (Fig. S15) suggesting that during BB influenced periods either higher NOX concentrations suppressed IEPOX and consequently IEPOX-SOA generation or the dominant BB aerosol during these periods inhibited measurement of isoprene oxidation products." This sentence does not have evidences to support. In theory, the high OM concentration from BBOA can facilitate the oxidation product to partition into the aerosol phase (Kroll and Seinfeld, 2008). To investigate the BB influence to IEPOX-SOA: 1) suggest the authors start to see if the BBOA plumes can increase the acidity of aerosol, which can promote the IEPOX uptake (Gaston et al., 2014). 2) the NOx concentration can influence the oxidation product of isoprene (Surratt et al., 2010; Bates et al., 2014). At the absent of IEPOX measurement, the authors can investigate the IEPOX-SOA concentration level or the fC5H6O abundance based on different categories of NOx concentrations by checking if there is systematical differences of IEPOX-

SOA mass concentrations at high NOx and low NOx. This analysis presumably assumed the isoprene emissions in the large areas were relatively constant within the field study period.

Minor comments

Page 4 line 1-2, Please give the standard deviations of those reported average values.

Page 4 line 27, How can the authors separate aged POA and SOA? To what extent, the authors will define the aerosol is aged POA since SOA can be from aged POA (Robinson et al., 2007).

Page 5 line 20, Have the authors offset the background of BAM measurement. e.g. the background concentration can refer to the mass concentrations of different species on blank filters.

Page 5 line 21, Suggest changing "PM1" to be "$PM_1$", which is more commonly used.

Page 6 line 17, Is the PTR-MS data UMR or HR? How did the authors separate the contributions of isoprene and furan to the m/z 69.

Page 7 line 14 Suggest adding the OM/OC values

Page 7 line 20, Please change "organic mass" to be OM. Since the authors start to use OM as an abbreviation, it should be consistent in the following text.

Page 7 line 25, Please give the standard deviations to those reported average concentrations.

Page 8 line 25. It is not clear that Org 60 is UMR or HR. The authors already use m/z 60 refers to UMR and $C_2H_4O_2^+$ refers to HR. Please be consistent. f60 (HR or UMR) has a similar problem. e.g., in page 9 line 1

Page 9 line 23, What does "atmospheric processing" refer to.

Page 11 line 4, Repeated sentence, which was already said in page 7 line 29.

Page 11 line 27, OOA should be SOA, OOA PMF factor has not been introduced here yet.

Page 11 line 8 "As these plumes were mostly related to sources close to the sampling site (within 2 to 10 km), differences in m/z 44 and 43 can be mostly related to different burning conditions." No evidence was shown to support this sentence.

For figures.

Figure 1. Suggest adding EC fraction in Fig. 1(a) to the total AMS fraction measured in Fig. 1(b), then a full chemical composition of $PM_1$ can be obtained.

Figure 3. Has the background of org 60 been subtracted (0.003). If the answer is negative, please offset the background.

Figure 4. Empirical triangle area was reported in Cubison et al. (2011). Suggest adding right guide line here as well, which can help to compare this study with other studies.

Figure 5-6. They are all the identical plots. It will be easier to see by combining them together as one panel figure.

Figure 9. (b) What is the correlation between BBOA with levoglucosan and $K^+$? what is the correlation between IEPOX-SOA with sulfate and isoprene? Does any compound correlate with OOA? Those external tracers should be added on the diurnal variation plot as well. Figure 9 (c): It is better to add the exact fraction values on the bar.

Figure 10. There are more f44 vs f82 points from IEPOX-SOA factors in Hu et al. (Hu et al., 2015). The authors should add the range of those data, which is more meaningful than showing only two points.

Reference:

Bates, K. H., Crounse, J. D., St. Clair, J. M., Bennett, N. B., Nguyen, T. B., Seinfeld, J. H., Stoltz, B. M., and Wennberg, P. O.: Gas Phase Production and Loss of Isoprene Epoxydiols, The Journal of Physical Chemistry A, 118, 1237-1246, 10.1021/jp4107958, 2014.

de Gouw, J. A., Middlebrook, A. M., Warneke, C., Goldan, P. D., Kuster, W. C., Roberts, J. M., Fehsenfeld, F. C., Worsnop, D. R., Canagaratna, M. R., Pszenny, A. A. P., Keene, W. C., Marchewka, M., Bertman, S. B., and Bates, T. S.: Budget of organic carbon in a polluted atmosphere: Results from the New England Air Quality Study in 2002, J Geophys Res-Atmos, 110, doi:10.1029/2004JD005623, Artn D16305

Doi 10.1029/2004jd005623, 2005.

Gaston, C. J., Riedel, T. P., Zhang, Z., Gold, A., Surratt, J. D., and Thornton, J. A.: Reactive Uptake of an Isoprene-Derived Epoxydiol to Submicron Aerosol Particles, Environ Sci Technol, 48, 11178–11186, 10.1021/es5034266, 2014.

Hayes, P. L., Ortega, A. M., Cubison, M. J., Froyd, K. D., Zhao, Y., Cliff, S. S., Hu, W. W., Toohey, D. W., Flynn, J. H., Lefer, B. L., Grossberg, N., Alvarez, S., Rappenglück, B., Taylor, J. W., Allan, J. D., Holloway, J. S., Gilman, J. B., Kuster, W. C., de Gouw, J. A., Massoli, P., Zhang, X., Liu, J., Weber, R. J., Corrigan, A. L., Russell, L. M., Isaacman, G., Worton, D. R., Kreisberg, N. M., Goldstein, A. H., Thalman, R., Waxman, E. M., Volkamer, R., Lin, Y. H., Surratt, J. D., Kleindienst, T. E., Offenberg, J. H., Dusanter, S., Griffith, S., Stevens, P. S., Brioude, J., Angevine, W. M., and Jimenez, J. L.: Organic aerosol composition and sources in Pasadena, California, during the 2010 CalNex campaign, Journal of Geophysical Research: Atmospheres, 118, 9233-9257, 10.1002/jgrd.50530, 2013.

Herndon, S. C., Onasch, T. B., Wood, E. C., Kroll, J. H., Canagaratna, M. R., Jayne, J. T., Zavala, M. A., Knighton, W. B., Mazzoleni, C., Dubey, M. K., Ulbrich, I. M., Jimenez, J. L., Seila, R., de Gouw, J. A., de Foy, B., Fast, J., Molina, L. T., Kolb, C. E., and Worsnop, D. R.: Correlation of secondary organic aerosol with odd oxygen in Mexico City, Geophys Res Lett, 35, L15804, Doi 10.1029/2008gl034058, 2008.

Hu, W. W., Campuzano-Jost, P., Palm, B. B., Day, D. A., Ortega, A. M., Hayes, P. L., Krechmer, J. E., Chen, Q., Kuwata, M., Liu, Y. J., de Sá, S. S., McKinney, K., Martin, S. T., Hu, M., Budisulistiorini, S. H., Riva, M., Surratt, J. D., St. Clair, J. M., Isaacman-Van Wertz, G., Yee, L. D., Goldstein, A. H., Carbone, S., Brito, J., Artaxo, P., de Gouw, J. A., Koss, A., Wisthaler, A., Mikoviny, T., Karl, T., Kaser, L., Jud, W., Hansel, A., Docherty, K. S., Alexander, M. L., Robinson, N. H., Coe, H., Allan, J. D., Canagaratna, M. R., Paulot, F., and Jimenez, J. L.: Characterization of a real-time tracer for isoprene epoxydiols-derived secondary organic aerosol (IEPOX-SOA) from aerosol mass spectrometer measurements, Atmos. Chem. Phys., 15, 11807-11833, 10.5194/acp-15-11807-2015, 2015.

Kroll, J. H., and Seinfeld, J. H.: Chemistry of secondary organic aerosol: Formation and evolution of low-volatility organics in the atmosphere, Atmos Environ, 42, 3593-3624, DOI 10.1016/j.atmosenv.2008.01.003, 2008.

Robinson, A. L., Donahue, N. M., Shrivastava, M. K., Weitkamp, E. A., Sage, A. M., Grieshop, A. P., Lane, T. E., Pierce, J. R., and Pandis, S. N.: Rethinking organic aerosols: Semivolatile emissions and photochemical aging, Science, 315, 1259-1262, DOI 10.1126/science.1133061, 2007.

Surratt, J. D., Chan, A. W. H., Eddingsaas, N. C., Chan, M., Loza, C. L., Kwan, A. J., Hersey, S. P., Flagan, R. C., Wennberg, P. O., and Seinfeld, J. H.: Reactive intermediates revealed in secondary organic aerosol formation from isoprene, Proceedings of the National Academy of Sciences, 107, 6640-6645, 10.1073/pnas.0911114107, 2010.

---

## Referee Comment (RC3) · Anonymous Referee #3 · 20 Sep 2016

This paper presents an analysis of atmospheric data from northern Australia, mainly AMS data impacted by biomass burning. A range of analyses is performed on the data, which gives some insight into the behaviour of the aerosol regarding the biomass burning plumes, ageing and also the formation of SOA from isoprene through the IEPOX route.

None of the results are particularly earth shattering or unexpected, given the pre-existing literature, and there is nothing really new on a process level compared to previous publications. However, there is currently a lack of in situ characterisation work like this in the tropics, so should probably be publishable on that basis. However, the paper is a little rambling and unfocused in paces, with discussions like isoprene SOA detracting from the supposed subject of the paper (ageing of biomass burning emissions), so the general theme of the paper should maybe be better defined.
[Figure]

I recommend publication for ACP, after the following comments have been considered:

Title: The title is possibly not appropriate, given that many other scientific phenomena other than the ageing of BB plumes (e.g. IEPOX-SOA) are discussed.

Page 8: 'Close BB' and 'distant BB' should be given more specific definitions. 'Prominent' is not very descriptive.

Page 10: Given the dynamic relationship between NOx and Ozone, it might be sensible to look at perturbations in 'potential ozone' (Ox = NO2 + O3) as well. This would remove the NO titration effect.

Page 10: The discussion of OA/CO vs time of day is problematic because the total amount of OA in a plume is the product of the total photochemical history of the airmass, not just the time of day that it was measured at. Without a more detailed treatment of the full airmass history, I do not see how any conclusions regarding things like fragmentation can be made.

Page 10: How is the change in SMPS size distributions consistent with SOA formation? Have you compared the particulate volume concentrations? Care should be exercised because an increase in mode diameter can occur simply through coagulation processes, which require no additional particulate mass formation.

Page 11: The assumption that f44 corresponds to photochemical activity is problematic. Biomass burning can produce a large amount of primary HULIS, which has a very high f44. Furthermore, while it has been shown that a plume's f44 will increase with time, it is not proven that photochemistry is necessarily responsible, particularly in the very early stages after emission where repartitioning or 'dark' chemical processes may occur. I would be more guarded and state that the high f44 implies a high level of oxygenation that could be caused by photochemistry.

Page 13: A lower boundary layer height can increase concentrations of primary emissions, but how would it increase IEPOX-SOA?

Page 14: There may be other explanations for a different f82, such as the formation of isoprene SOA through other routes (e.g. MPAN).

---

## Author Comment (AC1) · 13 Dec 2016

Author response to Referee #1

The authors thank the reviewer for interest in our work and for the helpful comments to improve the manuscript. We have addressed each of comments as outlined below. All page and line numbers refer to the revised manuscript (Revised_manuscript_trackChanges.docx) where all changes (track changes) are available (revised supplementary material is included in the revised manuscript pdf file). The link for the revised manuscript is given at the end of this document. If the text is significantly changed only the number of section is given (e.g. "Biomass burning events", Sect. 3.2.4).

Comment: This manuscript studied the biomass burning organic aerosol in Australia

by using a c-ToF-AMS. The heavy biomass burning causes that 90% of NR-PM1 is organics. Five intense BB events with different burning conditions are analyzed. The enhancement of ozone and SOA in biomass burning plumes in examined. PMF analysis on background OA resolved three factors, BBOA, OOA, and IEPOX-SOA. The IEPOX-SOA is identified for the first time in Australia. The manuscript is well structured and the content is appropriate for ACP. However, most of the work presented here has been done before. The manuscript in its current form does not add significantly to the literature. Thus, I recommend major revisions. More in depth analysis and conclusions are required.

Response: We agree that this kind of work has been done previously, however it is the first time that this has been done within the Australian continent. The Australian continent is globally an important source of BB emissions, particularly the tropical savannah region. However, data related to BB emissions are remarkably lacking for this region. Therefore, we think that the results from this work provide valuable information to enhance the understanding of the loading and composition of fresh and aged organic aerosols from biomass burning in Australia, Southern Hemisphere and in general.

The following has been modified or added to the manuscript: • The title of the manuscript is modified. • Separation on "close BB" and "distant BB" periods is clarified. • More analysis and discussion are added for BB events, including back-trajectories and Sentinel hot spot locations. • The suggestions regarding isoprene-derived OA are incorporated in the manuscript. • Discussion regarding the sea salt-chlorides and f44-resolved size distribution is modified and clarified.

Major comments

Comment 1: Whether this study is suitable/capable to study the aging of biomass burning OA. Previous field studies on this topic usually used aircraft to follow the biomass burning plumes, which ensures investigating the same air mass (Cubison et al., 2011; Yokelson et al., 2009; Forrister et al., 2015; May et al., 2015). However, this is not

the case in this study as the air mass is changing. In fact, many observations in this study can be explained by changes in air mass or physical changes, instead of aging of biomass burning OA. For example, an increase in f44 and a decrease in f60 (Figure 4b) could be simply due to an BBOA-rich air mass mixing with OOA-rich background air or physical changes during dilution (May et al., 2015). This could be potentially revealed in Figure 4a (close BB) if the authors color the data points by OA concentration. The data points with higher f60 and lower f44 would have higher OA concentration.

The authors may use MODIS sensor and backtrajectory analysis (section 2.1.7) to pinpoint fire position and calculate transport time. Then, the oxidation rate of BBOA or its tracers (such as f60) can be calculated, which is highly uncertain now. The aging time may also be estimated by using the photochemical clock, such as NOx/NOy ratio.

The OA in the five biomass burning events (section 3.2.4) is dominantly from fresh biomass burning, which provides a good opportunity to study the properties of fresh BBOA. The study would benefit from expanding related discussions. For example, "different burning conditions" are vaguely mentioned multiple times to explain the difference between the five biomass burning events. However, can the authors be more explicit about the relationship between specific burning conditions/burning materials with OA concentration, f44, f60, $\Delta$O3, etc? It is helpful to bring Table S2 to the main text.

Response 1: The authors are aware that more conclusions about the BB aging can be drawn when tracking the plume is available. However, the limitation of the stationary measurements made it extremely difficult to track any plume with certainty. We agree that process of aging is not covered by this study. However, results for aged BB emissions, including SOA and ozone formation and change in particle size, are presented in this study. Therefore, the title of the manuscript is changed from: "Aging of aerosols emitted from biomass burning in northern Australia" to "Fresh and aged aerosols emitted from biomass burning in northern Australia". The sentence at Page 3, lines 14-16 is also modified: "This publication presents insights into fresh and aged aerosols emitted during the SAFIRED, while a detailed description of the campaign can be found in Mallet et al. (2016)".

Considering the number of fires that occurred across the area during the campaign, ground-based (stationary) measurements and the limitations of satellite detection (once per day) it was difficult to identify the exact source of fires. The backtrajectories analysis was done for all days and all elevated signals, and while the path of trajectories can indicate the sources it is a challenge to apportion the signals to a specific event. In the case of all close fires, according to backtrajectories air masses could reach the ATARS within less than half an hour. The approximate time for air masses to reach the sampling location from the particular fire was included into the "Biomass burning events" section (Sect. 3.2.4) at Page 12.

The "biomass burning events" section (Sect. 3.2.4) is expanded as suggested. The Sentinel hotpots and the backtrajectories for BB events are presented. More discussion about differences in BB events is given. Text was changed through the "Biomass burning events" section (Sect. 3.2.4) at Page 12. Table S2 is now in the main text (Table 1).

Comment 2: IEPOX-SOA related. Firstly, it is not appropriate to label the PMF factor as IEPOX-OA. While the majority of this factor is from the reactive uptake of IEPOX, this factor also likely includes contribution from isoprene OA formed via other pathways. For example, a recent study by Schwantes et al. (2015) showed clearly that SOA formed via the reactive uptake of isoprene nitrooxy hydroxyepoxide could also produce fragment m/z 82. In addition, the mass concentration of this PMF factor cannot be fully explained by the total IEPOX SOA tracers (Hu et al., 2015; Budisulistiorini et al., 2015). Thus, it is more appropriate the name the factor as isoprene-derived OA (Xu et al., 2016; Pye et al., 2016). Secondly, since the isoprene-OA factor is identified for the first time in Australia, more analysis could be done related to this factor. For example, does this factor correlate with sulfate, which has been shown in previous studies (Xu et al., 2015a; Budisulistiorini et al., 2015)? What's the NO concentration at the measurement

site? Is IEPOX formed locally or from transport? What's the particle pH? As shown in Figure 9f, in general, IEPOX-OA correlates well with Org 82. However, some data are scattered (i.e., Org 82 0.004 to 0.008 $\mu$g m-3 range). What happens for these data? Is it due to sources other than IEPOX uptake contributing to Org 82?

Response 2: Thank you for the comment. The PMF factor "IEPOX-SOA" is changed for "isoprene-derived OA" throughout the manuscript and supplementary material. Moreover, the following sentence is added to Page 15, lines 24-26: "In order to consider other possible pathways in isoprene-derived OA formation (Schwantes et al., 2015), the PMF factor has been named as isoprene-derived OA (Pye et al.; Xu et al., 2016)".

According to calculations for the composition depended collection efficiency, 22% of the aerosol were acidic. Furthermore, the correlation of IEPOX-SOA and sulfate is found to be weak (R of 0.3) (Fig. S18). However, two periods (period before 5th of June and after 15th of June) could be clearly separated on the plot. While there is no correlation between sulfate and isoprene-derived OA factor for the first period, when plotting only data collected from the 15th of June, correlation is found to be slightly higher (R of 0.4). The following is added to the manuscript at Page 16 line 6-14: "The main path for isoprene-derived OA formation is proposed to be acid catalysed IEPOX uptake (Lambe et al., 2015; Lin et al., 2011). According to calculations for the composition depended collection efficiency, 22% of the aerosol were acidic. In order to estimate whether the acidity of the particles had an influence on isoprene-derived OA generation in ATARS, the correlation between sulfate (taken as proxy of aerosol acidity) and isoprene-derived OA factor was examined. The correlation between the factor and sulfate can be considered as weak (R of 0.3) (Fig S18). However, two periods can be clearly distinguished from the graph: the period before 5th of June and the period after 15th of June. While there is no correlation between sulfate and isoprene-derived OA factor for the first period, when plotting only data collected from 15th of June, correlation is found to be slightly higher than correlation for all background data (R of 0.4)".

The concentration of NO at the measurement site is not available. Unfortunately due to

the issue with instrumentation, NOx data are unreliable and thus not used in analysis.

As isoprene-derived OA was found to be prominent over the night period, one of the suggestions was detection of distant masses as suggested at Page 16, line 27.

After further investigation of the scattered data (i.e., Org 82 0.004 to 0.008 $\mu$g m-3 range) we couldn't tell what other source contributed. Further analysis of the biogenic loadings and sources for this region are needed in order to make more detailed conclusions regarding the isoprene-derived OA.

Minor comments

Comment 1: Page 2, line 6: Elemental carbon (EC) is not the same as black carbon (BC). EC is defined by thermal properties and BC is defined by optical properties (Andreae and Gelencsér, 2006).

Response 1: Elemental carbon (EC) is used throughout the manuscript. "Black" is removed from the sentence at Page 2, line 6.

Comment 2: Page 4, line 4. It is stated that sampled air mass had mainly passed over land. However, this seems to contradict with the conclusion in Page 8 Line 18-20, that a significant portion of chlorides could be sea salt.

Response 2: The sampled air masses mainly passed over the land according to a dominant wind direction (southeast) and backtrajectories, as illustrated in the Figure 1 in Mallet et al. (2016). The high chloride concentration was detected (by the AMS, Figure 1c and by the BAM Figure S3a) during the close BB periods when air masses were passing over the land affected by fires (as stated in the manuscript at Page 8 line 27), meaning that they originated from fires. This is especially the case of close fires on the 25th and 26th of June (Fig. 1c). However, oceanic influence cannot be excluded as on some days (e.g. period between 3rd and 6th of June) in the afternoons northeast winds were dominant (Figure 4c in Mallet et al. (2016)). More precisely on the way to the sampling station air masses were passing over the land and then

turning over the ocean before reaching the ATARS. Furthermore, the measurement period between 19th and 22nd of June was characterised by air masses coming mostly along the Australian east coast and passing over some of the land before reaching the station (Mallet et al., 2016). The oceanic influence through high chloride concentration that has a sea salt origin can be clearly seen from the Fig S3a. Chlorides from BAM were detected during the influence of close BBs, but also during the oceanic influenced days. Therefore the Mg2+/Na+ ratio for oceanic-influenced days was close to the see salt ratio (Figure S3b) and the sea salt-related chlorides were collected on filters (BAM) (Figure S3a). The least terrestrial fetch (indicated by low radon concentration) was also observed during the days with the high ocean breeze (Mallet et al., 2016).

To clarify this in the manuscript the following is added at Page 4, lines 4-7: "However, on some days (e.g. between 3rd and 6th of June) in the afternoon hours northeast wind direction was dominant, directing the air masses from land to pass over the ocean before being detected at ATARS. Moreover, an oceanic influence was observed between 19th and 22nd of June (Mallet et al., 2016)".

The following sentences are modified on the Page 9, line 3 – Page 9 line 8: "Mg2+/Na+ ratio values for the filters collected during the high oceanic influence (between 3rd and 6th and 19th and 22nd of June) were close to the sea salt Mg2+/Na+ ratio of 0.12 (Fig. S3b). At the same time low terrestrial fetch (low radon concentration) was observed (Mallet et al., 2016). Moreover, the chlorides collected on filters were prominent during the period of oceanic influence (Fig. S3a). This suggests that a significant portion of chlorides detected on the BAM filters was of a sea salt origin, which is refractory and therefore not well measured by the AMS, thereby explaining the poor chloride correlation" and at Page 8, line 27: "Increased chloride concentration during the close BB periods was also indicated by the BAM data (Fig S3a)".

A new Figure, Fig S3a that shows the BAM chloride time series is added to the supplementary material and Fig. S3 is now Fig. 3b. Figure 3b is also modified: z-axis (date) was added to the graph.

Comment 3: Page 4 Line 19-20. It is not clear if the CE is determined by using the UMR data or the HR data. Or HR data are used in the squirrel panel? Also, sulfate fragments should not have interference from organic fragments in HR data. Thus, HR data should be used throughout the analysis.

Response 3: CE was determined using the HR data in PIKA and these data are used through the analysis as stated now at Page 4, lines 21 and 22: The AMS collection efficiency was determined using the calculations provided within the PIKA Toolkit" and Page 4, lines 25-27: "Significant improvement was made in distinguishing sulfate fragments from organic fragments at the same m/z by performing HR peak fits in PIKA (Sect. S1 in supplementary material). Therefore, HR peak fitting fata (PIKA) were chosen for further analysis".

Comment 4: Page 7, line 31: It is not clear how the "close BB" and "distant BB" periods are separated. The criteria for separation should be clearly stated?

Response 4: The authors agree that more specific definitions should be given for "close" and "distant" BB periods and clear presentation of data separation should be given.

The measurement period was separated into "close BB" and "distant BB" periods based on organics, carbon monoxide and particle number concentration and their correlation, as well based on the distance of known fires (Sentinel hot spots) from the ATARS.

Firstly, we were looking at the time series of organics, carbon monoxide and particle number concentration (Figure 2, Page 25) in order to identify the periods of significant BB influences, e.g. their high signals (e.g. on 9th and 26th of June) gave us a guide on which days intense/close BB took place. The increases in BB markers (levoglucosan, soluble non-sea salt potassium and Org 60) also showed the same trend (Figure 3, Page 26). After identifying these periods, the next step was to relate these signals to the possible BB events. Therefore, we have mapped all fires (Sentinel hot spots) that

have happened during these periods and have confidence level of 50% and more. To make sure that everything is included (e.g. close fires with low intensity) we have gone through every measurement day and mapped all the events that have occurred.

Hot spots detected within 20 km occurred on 30th of May, between 7th and 11th and 25th and 26th of June (Figure 2, Page 25). Taken the distance of 20 km and increased concentrations of organics, carbon monoxide and particle number, these days were taken as "close BB" periods. Knowing that ATARS was constantly affected by the BB air masses the remaining measurement days were taken as "distant BB" periods. In summary, "close BB" periods refer to periods where close fires (within 20 km) have taken the place and correlated increases in carbon monoxide, organics and particle number concentration were observed (Fig. 2), while the rest of the measurement is defined as "distant BB" periods.

In order to clarify the close and distant BB periods the paragraph at Page 8, line 3- Page 8 Line 16 is modified: "The sampling site was constantly impacted by fire emissions with periods of heavy BBs characterized by high aerosol and gas phase concentrations, for instance CO reaching up to $\sim$104 ppb and organics up to 350 $\mu$g m-3 (Fig. 2). The most intense BB episodes were on the 30th of May, between the 7th and 11th of June, and on the 25th and 26th of June (Fig. 1c) (Mallet et al., 2016). During intense fire periods, organics, CO and particle number concentration showed correlated increases (Fig. 2). Moreover, looking at Sentinel Hot spots during these periods, hot spots were detected within 20 km from the ATARS. Based on this, the dataset was separated into periods of "close BB" (corresponding to high organics, CO and particle number concentration signals and close events (< 20 km)) and "distant BB" (corresponding to less intense organics, CO and particle number concentration signals and distant events (> 20 km)). It is important to emphasize that all periods during the measurement that have not been included in close BB periods have been assigned to distant BB periods, as ATARS was constantly influenced by BBs. The selection does not mean that emissions from distant fires were not present during the close BB periods but that the

influence of fires near the measurement station during these periods was dominant. Nine intense BB events were identified from close BB periods as single source emissions (Desservettaz et al., 2016). Five of the nine BB events (Fig. 2) were analysed here (Sect. 3.2.4), due to the instrument not sampling during the remaining events. Most of the events occurred in the afternoon/night time (Table 1)".

In order to indicate distant BB periods the following is added to Fig. 2 caption at Page 25: "The distant BB periods cover all days of the measurement other than days included in close BB periods".

Comment 5: Page 8, line 14: The authors can use the NO+/NO2+ ratio to infer if the nitrate functionality originates from inorganic or organic nitrate (Xu et al., 2015b; Farmer et al., 2010).

Response 5: According to the NO+/NO2+ ratio in our study, close BB period between 7th and 11th of June is characterised by nitrate that originates from the ammonium nitrate rather than from organic nitrate (NO+/NO2+ ratio is higher for organic nitrates than ammonium nitrate (Farmer et al., 2010)) (see Figure below). Other two close BB periods (on 30th and between 25th and 26th) produced inorganic as well as organic nitrates. Periods influenced with ocean breezes (between 3rd and 5th and 19th and 22nd of June) seem to have higher portion of organic than inorganic nitrates.

Time series of NO+/NO2+ ratio throughout the campaign. Solid lines indicate close BB periods. Dashed line indicates NO+/NO2+ ratio for ammonium nitrate.

Comment 6: Page 10, line 22-23: This sentence is really confusing and should be re-written.

Response 6: The diurnal trends of the f44, △O3/△CO and △OA/△CO ratios for close BB periods and diurnal trend of △O3/△CO ratio for distant BB periods are added to the Fig. 5 and therefore apart from the sentence that was mentioned in the comment, the changes are made throughout the "Secondary organic aerosol (SOA) formation"

[Figure]

section (Sect. 3.2.2 at Page 11).

Comment 7: Page 11, line 22. Is the OA mass loading related to wind speed or wind direction?

Response 7: The wind direction was highly variable influencing detected signals related to the event F. This is incorporated into the manuscript at Page 13, lines 22-26: "However, the f44 value was highly variable during the event F, ranging from 0.07 to 0.18. Other parameters including f60 and organics varied as well. On that time, the wind direction significantly varied between 140° and 80° and likely influenced changes in detected air mass. One of the explanations can be detection of fresh plume and aged masses coming from the distant fires with the change in wind. The low correlation between f60 and f73 might also be an indicator for detection of different BB air masses (Fig. S8)".

Comment 8: Page 12, line 14. The authors may check if the relationship between f60 and f73 changes with burning conditions or burning materials.

Response 8: We have observed changes in f60/f73ratio for different events (varying between 1.2 and 2.5) (Fig. S8), however there was no trend in relationship of f60/f73 ratio and burning conditions/material (using the MCE value). Low correlation between f60 and f73 for the event F (r2=0.16) was likely due to the changes in masses that were detected during this time period (due to high variability in wind direction).

The sentence at Page 12, lines 16 and 17 is modified: "Different factors were considered including f44, time of day, $\Delta O3/\Delta CO$, f60, f60/f73, organic concentration and MCE". The following is added to the manuscript at Page 13 lines 25 and 26: "The low correlation between f60 and f73 might also be an indicator for detection of different BB air masses (Fig. S8)" and at Page 14, lines 19-21: "Events were characterised with different f60/f73 ratios (varying between 1.2 and 2.5), but there was no trend in the relationship of f60/f73 ratio and burning material/conditions (MCE values) (Fig. S8)".

[Figure]

Figure illustrating f60/f73 ratio changes for different events and relationship between this ratio and MCE (Fig. S8) is added to supplementary material.

Comment 9: Page 12, line 31. Is there any study to show that fresh emissions from biomass burning contain OOA?

Response 9: Contribution of primary (fresh emissions) from biomass burning to OOA portion of emitted OA cannot be excluded as shown in wood burning studies, Heringa et al. (2012) and Weimer et al. (2008). The references are added at Page 15, lines 12 and 13.

Comment 10: Page 13, line 10. Cite Xu et al. (2015a).

Response 10: The citation is added to the new version of the manuscript at Page 15, line 23.

Comment 11: Page 14, lines 8-11: The low correlation between IEPOX-OA and BBOA could be simply due to that these two factors are from completely different sources and a correlation is not expected. The discussion in this sentence is not supported and should be removed.

Response 11: The sentence was removed from the manuscript (Page 17, lines 11-14) and figure S15 (supplementary material) that corresponds to the statement was removed as well.

Heringa M, DeCarlo P, Chirico R, Lauber A, Doberer A, Good J, et al. Time-resolved characterization of primary emissions from residential wood combustion appliances. Environmental science & technology 2012; 46: 11418-11425. Mallet MD, Desservettaz MJ, Miljevic B, Milic A, Ristovski ZD, Alroe J, et al. Biomass burning emissions in north Australia during the early dry season: an overview of the 2014 SAFIRED campaign. Phys. Discuss., doi:10.5194/acp-2016-866, in review, 2016 2016. Pye HO, Murphy BN, Xu L, Ng NL, Carlton AG, Guo H, et al. On the implications of aerosol liquid water and phase separation for organic aerosol mass. Schwantes RH, Teng AP, Nguyen

[Figure]

TB, Coggon MM, Crounse JD, St. Clair JM, et al. Isoprene NO3 Oxidation Products from the RO2+ HO2 Pathway. The Journal of Physical Chemistry A 2015; 119: 10158-10171. Weimer S, Alfarra M, Schreiber D, Mohr M, Prévôt A, Baltensperger U. Organic aerosol mass spectral signatures from wood‐burning emissions: Influence of burning conditions and wood type. Journal of Geophysical Research: Atmospheres (1984–2012) 2008; 113. Xu L, Middlebrook AM, Liao J, Gouw JA, Guo H, Weber RJ, et al. Enhanced formation of isoprene‐derived organic aerosol in sulfur‐rich power plant plumes during Southeast Nexus. Journal of Geophysical Research: Atmospheres 2016; 121.

Please also note the supplement to this comment:
http://www.atmos-chem-phys-discuss.net/acp-2016-730/acp-2016-730-AC1-supplement.pdf

---

## Author Comment (AC2) · 13 Dec 2016

Author response to Referee #2

The authors appreciate the reviewer's comments that helped us to improve the manuscript. We have addressed each of comments as outlined below. All page and line numbers refer to the revised manuscript (Revised_manuscript_trackChanges.docx) where all changes (track changes) are available (revised supplementary material is included in the revised manuscript pdf file). The link for the revised manuscript is given at the end of this document. If the text is significantly changed only the number of section is given (e.g. "Biomass burning events", Sect. 3.2.4).

Comment: This paper discussed the aging of biomass burning (BB) aerosols from

[Figure]

Northern territory of Australia based on aerosol mass spectrometer (AMS) at a ground observation site. The entire field study (29 May to 20 June, 2014) was divided to be "close BB" and "distant BB" periods. The BB aerosols were investigated by comparing aerosol chemical composition, aging process, ozone formation, size distribution and SOA formation between those two periods. PMF analysis was performed on the background OA period. Ambient Isoprene Epoxydiols-Derived SOA (IEPOX-SOA) factor from three PMF-factors solution was discussed and showed similar performances to the same factor reported in the previous studies. This paper showed a good dataset for investigating the biomass burning influences on aerosols. However, the analysis in this paper is not clear and is not presented in a quite logic way. Thus I recommend a major revision for this paper before considering its publication in ACP.

Response: Number of changes is made throughout the manuscript in order to present data in more clear way. The following has been modified or added to the manuscript: The title of the manuscript is modified. Separation on "close BB" and "distant BB" periods is clarified. More analysis and discussion are added for BB events, including backtrajectories and Sentinel hot spot locations. The suggestions regarding isoprene-derived OA are incorporated in the manuscript. Discussion regarding the sea salt-chlorides and f44-resolved size distribution is modified and clarified.

Comment 1: The authors analysed the data in section 3.2 based on dividing the entire study to be "distant BB" and "close BB" periods in section 3.1. However, it is not clear that how the "close BB" and "distant BB" periods were defined. Was it based on distance or the emission intensity, or the correlation between organic, CO and particle number concentration? The authors only labelled "close BB" period in Figs.1-2, "distant BB" period and background OA period are unknown.

Response 1: The authors agree that more specific definitions should be given for "close" and "distant" BB periods and clear presentation of data separation should be given.

[Figure]
The measurement period was separated into "close BB" and "distant BB" periods based on organics, carbon monoxide and particle number concentration and their correlation, as well based on the distance of known fires (Sentinel hot spots) from the ATARS.

Firstly, we were looking at the time series of organics, carbon monoxide and particle number concentration (Figure 2, Page 25) in order to identify the periods of significant BB influences, e.g. their high signals (e.g. on 9th and 26th of June) gave us a guide on which days intense/close BB took place. The increases in BB markers (levoglucosan, soluble non-sea salt potassium and Org 60) also showed the same trend (Figure 3, Page 26). After identifying these periods, the next step was to relate these signals to the possible BB events. Therefore, we have mapped all fires (Sentinel hot spots) that have happened during these periods and have confidence level of 50% and more. To make sure that everything is included (e.g. close fires with low intensity) we have gone through every measurement day and mapped all the events that have occurred.

Hot spots detected within 20 km occurred on 30th of May, between 7th and 11th and 25th and 26th of June (Figure 2, Page 25). Taken the distance of 20 km and increased concentrations of organics, carbon monoxide and particle number, these days were taken as "close BB" periods. Knowing that ATARS was constantly affected by the BB air masses the remaining measurement days were taken as "distant BB" periods. In summary, "close BB" periods refer to periods where close fires (within 20 km) have taken the place and correlated increases in carbon monoxide, organics and particle number concentration were observed (Fig. 2), while the rest of the measurement is defined as "distant BB" periods.

In order to clarify the close and distant BB periods the paragraph at Page 8, line 3- Page 8 Line 16 is modified: "The sampling site was constantly impacted by fire emissions with periods of heavy BBs characterized by high aerosol and gas phase concentrations, for instance CO reaching up to $\sim$104 ppb and organics up to 350 $\mu$g m-3 (Fig. 2). The most intense BB episodes were on the 30th of May, between the 7th and 11th of June,

and on the 25th and 26th of June (Fig. 1c) (Mallet et al., 2016). During intense fire periods, organics, CO and particle number concentration showed correlated increases (Fig. 2). Moreover, looking at Sentinel Hot spots during these periods, hot spots were detected within 20 km from the ATARS. Based on this, the dataset was separated into periods of "close BB" (corresponding to high organics, CO and particle number concentration signals and close events (< 20 km)) and "distant BB" (corresponding to less intense organics, CO and particle number concentration signals and distant events (> 20 km)). It is important to emphasize that all periods during the measurement that have not been included in close BB periods have been assigned to distant BB periods, as ATARS was constantly influenced by BBs. The selection does not mean that emissions from distant fires were not present during the close BB periods but that the influence of fires near the measurement station during these periods was dominant. Nine intense BB events were identified from close BB periods as single source emissions (Desservettaz et al., 2016). Five of the nine BB events (Fig. 2) were analysed here (Sect. 3.2.4), due to the instrument not sampling during the remaining events. Most of the events occurred in the afternoon/night time (Table 1)".

In order to indicate distant BB periods the following is added to Fig. 2 caption at Page 25: "The distant BB periods cover all days of the measurement other than days included in close BB periods".

Background period is related just to the PMF analysis and therefore it was not introduced in the 3.1 chapter, but it is clearly defined at Page 15, lines 1 – 4 and illustrated in Figures S13 and S14. Comment 2: Suggest revising the structure of the results and discussion part. After all comparisons between the "close BB" and "distant BB" being discussed in section 3.2.1-3.2.3, the paper started to describe the details about the "close BB" period in section 3.2.4, which is very confusing. I do not quite understand what the relationship between the 5 BB events in section 3.2.4 with the "close BB" period analysed in section 3.2.1-3.2.3 is. I suggest the authors introduce the details on "close BB" and "distant BB" period first, and then discuss the general comparison results. In this way, the readers can get a rough idea on what "close BB" and "distant BB" are then understand later comparison better. I also suggest the authors add a separate figure for readers to see the details of those "close BB" and "distant BB" periods. In this way, the good correlation addressed in page 7 Line 30 can also be understood.

Response 2: As suggested (previous comment) the "close BB" and "distant BB" periods were clearly defined and introduced before the comparison of the results. Moreover the following is added to the manuscript at Page 8, lines 14-16: "Nine intense BB events were identified from close BB periods as single source emissions (Desservettaz et al., 2016). Five of the nine events (Fig. 2) were analysed here (Sect. 3.2.4), due to the instrument not sampling during the remaining events". This is repeated at the beginning of the "Biomass burning events" section (Sect.3.2.4) at Page 12, lines 10-12: "Five single source BB events were analysed here (Fig. 2). These episodes were within previously defined close BB periods". This clearly explains that nine events were identified as a single source events (coming just from the one fire source) and that these events are within the close BB periods. The AMS was not sampling during all events, therefore, five events labelled on Fig. 2 are addressed in this paper. As discussed in the previous comment, distant BB periods were clarified in the main text and in the figure caption (Fig. 2, Page 25).

Comment 3: Page 8, line 10-20: 1) The authors mentioned there were high chloride (KCl or NH4Cl) mass concentrations from the biomass burning, then said a significant portion of chlorides from BAM measurement came from sea salt. Those two statements are conflictive.

2) If the signals are strong, KCl+ and NaCl+ ions can be resolved in the OA spectra in the HR fitting of PIKA.

3) The chloride with high concentrations in Fig.1 was not shown in the comparison plot of Fig. S2. Were some periods missed in Fig. S2? The authors should point out which periods were used for comparison in Fig.S2.

4) Line 10, "This can be explained by the depletion of chloride with transport and aging of BB plumes (Li et al., 2003; Li et al., 2010;Liu et al., 2000)." There is no proof for this sentence. Since the author has the K+ and Na+ measurement, the authors can calculate the chloride depletion fraction to show the evidence for this sentence. See equation (A1) and Fig. (A2) in (Hayes et al., 2013) for reference.

Response 3: 1) High AMS-resolved chloride concentrations during the close BB periods can be clearly seen from the AMS (Fig.1(c)) and BAM (Fig. S3a) time series and also high correlation of AMS-resolved chlorides and organics during the close BB periods indicates BB-related origin. The high contribution of chlorides originating from fires has been also emphasised in the campaign overview publication (Mallet et al., 2016). However, apart from the fire-related chloride signals, BAM filters collected the sea salt chlorides during the periods of oceanic influence (period between 3rd and 6th and 19th and 22nd of June, as it can be seen in the Fig S3a. Moreover, the Mg2+/Na+ ratio (ions measured by BAM) for the filters collected during the oceanic-influenced periods was close to the ratio for the sea salt (Fig. S3b). Therefore we concluded that significant portion of chlorides from BAM filters were from the sea salt. For more details please see the response to the minor comment number 2, from Referee 1.

To clarify this in the manuscript the following is modified to the Page 9, line 3- Page 9, line 8: "Mg2+/Na+ ratio values for the filters collected during the high oceanic influence (between 3rd and 6th and 19th and 22nd of June) were close to the sea salt Mg2+/Na+ ratio of 0.12 (Fig. S3b). At the same time low terrestrial fetch (low radon concentration) was observed (Mallet et al., 2016). Moreover, the chlorides concentration collected on filters was prominent during the period of oceanic influence (Fig. S3a). This suggests that a significant portion of chlorides detected on the BAM filters was of a sea salt origin, which is refractory and therefore not well measured by the AMS, thereby explaining the poor chloride correlation". Moreover, the sentence was added at Page 8, line 27: "Increased chloride concentration during the close BB periods was also indicated by the BAM data (Fig S3a)".

In order to present the BAM chloride time series a new Figure, Fig S3a is added to the supplementary material and Fig. 3 is now to Fig. 3b. Figure 3b is also modified: z-axis (date) was added to the graph.

2) After HR analysis of KCl+ and NaCl+ ions, further analysis couldn't be done as resolving the ions was not possible due to their low contribution (see in Figure below).

3) All AMS data were averaged to BAM 12h data, therefore the AMS concentrations at Fig.1 and Fig. S2 are not the same. To emphasise this, the following sentence is added to the Fig. S2 caption: "The AMS data were averaged to BAM 12h data".

4) Unfortunately, we were not able to track the plumes and it was a challenge to separate the plumes and investigate their aging and thus chlorides depletion. Therefore, further analysis could not be performed.

Comment 4: Page 9, line 7: "The variability observed in f44 vs. f60 for close BB events (Fig. 4a) probably reflect BB plumes generated during different burning conditions rather than different atmospheric processing of BB masses." There is no evidence to support this sentence. In Fig. 5a, higher $\Delta O3/\Delta CO$ was observed for the high f44. The authors explained this figure as (page 9 line 31): "On average, the $\Delta O3/\Delta CO$ ratio increases with f44 and decreases with f60, indicating increased photochemical processing of OA in plumes with atmospheric aging and ozone production". Two statements are contradicting with each other.

Response 4: The authors have changed the sentences at Page 9, lines 25-27 as following: "The variability observed in f44 vs. f60 for close BB events (Fig. 4a) can reflect BB plumes generated during different burning conditions but also different atmospheric processing of BB masses" Ozone enrichments were observed in close as well as in distant plumes, therefore this statement is kept as it is and in addition sentence is added at Page 10, lines 28 and 29: "The $\Delta O3/\Delta CO$ enrichments for close BB period indicate that aging of close emissions cannot be excluded".

Comment 5: Page 9, line 14: "This confirms that levoglucosan-like species carried by the BB plume did not degrade to background levels even as oxidized species were formed. Thus, f60 is a reasonable marker of distant BB in this study." This expression is not quite true; lots of points in Fig 4b are around background level when the f44 is very high. Mixing BB plumes with other aged plumes also can degrade the f60.

Response 5: Most of the points in Fig. 4b are higher than the f60 background level when f44 is high (from 0.2 to 0.25). Mixing of BB plumes can result in lower f60, but even then most of the points are higher than the background. So the overall conclusion is that levoglucosan can be used as a reliable marker for distant BBs in this study.

Comment 6: Page 9 line 23 "The wide range in f44 can be attributed to difference in burning conditions for close BB". What the author missing in this entire section is dilution or mixing BB plumes with other plumes. Not only different burning condition or aging, but also the dilution and mixing will also lead the similar evolution trend of f44 vs f60 in the triangle plot (Fig. 4).

Response 6: Dilution effect: The authors agree that dilution effect should be included in the text, as a factor influencing f44 vs f60 trend. This is added to the manuscript at Page 10, lines 11 and 12: "One more factor that can influence f44 vs f60 trend for both close and distant fire is dilution effect".

Mixing between plumes: The backtrajectories clearly show that during the close BB periods, apart from the close fires all other hot spots are located on 60km or more from the ATARS (Fig S7 in the supplementary material). The authors thus suggest that the mixing of fresh plumes with the plumes coming from the distant fires in most of the cases won't influence the f44/f60 changes due to the fact that both m/z 44 (60) and total organics for distant diluted masses are likely much smaller than m/z 44 (60) and total organics for close plume masses. This is simplified through the equation below.

f44=(Org 44 (fresh plume masses)+Org 44 (diluted distant plume masses))/(Org (fresh plume masses)+Org (diluted distant plume masses))

As same implies for f44 and f60, it is assumed that the mixing of fresh plumes with the distant diluted plumes coming from the distant fires won't affect the f44/f60 trend.

As not the same can be concluded in case for the distant BB periods the following is added to the manuscript at Page 10, line 7 and 8: "The observed evolution trend in f44 vs f43 for distant fires can be also influenced by mixing between the plumes" and sentence at Page 14, lines 26-28 is modified: "The thousands of fires that occurred during the SAFIRED campaign contributed to a wide range of OA composition that reflect different burning materials, conditions, mixed fresh and aged emissions and processing in the atmosphere".

Comment 7: Page 10, line 2: Suggest the authors use Ox (=NO2+O3) instead of O3, or use both, then the O3 loss by NO can be accounted for (Herndon et al., 2008).

Response 7: This comment unfortunately cannot be addressed due to the unreliable NOx data.

Comment 8: Page 10, lines 6 and 7: How did the authors calculate the average $\Delta O3/\Delta CO$ from different fires? How did the authors obtain $\Delta O3/\Delta CO$ from "distant BB" fire plumes since the plume spikes are not obvious as those in "close BB" period?

Response 8: The average $\Delta O3/\Delta CO$ ratio for particular BB event (5 events that were singled out) was calculated by averaging all values for the time of the fire. The average $\Delta O3/\Delta CO$ ratio for distant BB periods is the average of all $\Delta O3/\Delta CO$ values for the whole distant BB periods, not for the fires in periods separately.

Comment 9: Page 10, line 14: The authors should calculate $\Delta OA/\Delta CO$ vs $\Delta f44$ (not f44), because it is not clear if high f44 can really reflect the SOA aging since different BB plumes may have different f44 due to burning condition and mixing or dilution. I suggest the authors check the photochemical age calculated from VOCs (de Gouw et al., 2005), which can be another parameter for indicating oxidation/aging process of plumes. And also the authors have a great dataset to see emission ratio of BB emitted

aerosols (e.g. $\Delta SO4/\Delta CO$, $\Delta NH4/\Delta CO$ and $\Delta Cl/\Delta CO$).

Response 9: The authors think that considering $\Delta f44$ (the change in f44 for different burning conditions) would only be beneficial for individual plumes, but not for the whole data set as we have presented here. Furthermore, for individual plumes, this is only possible if the plume can be tracked and the initial value of f44 is known. That is not the case for this study (stationary measurement site).

Unfortunately, due to the numerous fires which occurred across the area and the limitations of satellite (passing the region once per day) and ground-based measurements, we were not able to single out individual plumes that could be used for the suggested analysis.

Comment 10: Page 10, 3.2.3 section: Suggest adding size distributions of m/z 44 and m/z 60, which can be used to compare with the size distributions from SMPS. This comparison can help to interpret the comparison between "low f44" and "high f44" periods. In addition, the comparison of size distributions between "close BB" and "distant BB" can be added. This can help to characterize those two periods and see if the "distant BB" are really more aged or mixed with aged aerosols.

Response 10: In order to further analyse changes in size distribution with aging, size distributions for organics (sampled by the AMS) were examined for both close and distant BB periods. Data were categorised based on different f44 ranges, same as for the SMPS data (Fig 6).

More conclusions were made and paragraph for the Sect. 3.2.3 at Page 11 is modified: "Atmospheric aging of plume particles increases particle diameter due to gas to particle transfer of organic and inorganic gaseous species (Martins et al., 1998). In order to estimate whether the aging of BB masses during SAFIRED influenced particle size, average SMPS size distributions and AMS size distributions for organics, both categorised based on different f44 ranges, were examined (Fig. 6). It is important to emphasise that SMPS uses electrical mobility diameter, while AMS uses vacuum aerodynamic diameter. The close and distant BB periods were analysed separately. The f44 values were classified into four groups that represent different aging stages ($0.05 < f44 < 0.1$, $0.1 < f44 < 0.15$, $0.15 < f44 < 0.2$, $0.2 < f44 < 0.25$). The first f44 bin ($0.05 < f44 < 0.1$) was not considered in case of the distant BB periods, as only a few data points were in this range (Fig. 4). The same was done for the highest f44 bin ($0.2 < f44 < 0.25$) for close BB periods. According to SMPS data, the average particle mode varied between 101 - 113 nm and 104 - 106 nm for close and distant BB periods, respectively. The average mode for organics showed larger sizes and varied between 259 - 293 nm and 293 - 305 nm for close and distant BB periods, respectively. Increased f44 was accompanied by a reduction in SMPS particle size for close plumes, going from 113 nm ($0.05 < f44 < 0.1$) to 101 nm ($0.1 < f44 < 0.15$). The same trend was observed for the organic aerosols. Considering both AMS and SMPS data for distant fires, there was no considerable change in diameter with aging from less aged BB plumes to more aged BB air masses. The particle modes show only slight differences between different f44 bins. This is not consistent with the observed increase in OA for distant fires. Changes in size distribution will be discussed further in the Sect. 3.2.4 where results related to specific BB events are presented".

Please note that after further analyses we realised that only a few points were in the f44 range 0.05-0.1 for distant fires, therefore this was removed from the manuscript and new graph was added (Fig. 6).

Comment 11: Page 14, lines 1-2: What is the monoterpene concentration in this study? Monoterpene-derived SOA also can influence the background level of $C_5H_6O^+$ (Hu et al., 2015). Isoprene concentration was not reported in this study, which is important for discussing IEPOX-SOA.

Response 11: The average monoterpenes concentration was found to be $0.22 \pm 0.41$ ppb. The average isoprene concentration was $0.49 \pm 0.78$ ppb. The following is added to the manuscript at Page 16, line 2-5: "The average isoprene concentration measured during the campaign was $0.49 \pm 0.78$ ppb. However, it should be emphasised that

monoterpene-derived OA can also influence the background level of C5H6O+ (Hu et al., 2015). The average monoterpenes concentration for this study was found to be $0.22 \pm 0.41$ ppb".

Comment 12: Page 14, lines 8-9: Low correlation between the IEPOX-SOA time series with both OOA and BBOA was observed (Fig. S15) suggesting that during BB influenced periods either higher NOX concentrations suppressed IEPOX and consequently IEPOX-SOA generation or the dominant BB aerosol during these periods inhibited measurement of isoprene oxidation products." This sentence does not have evidences to support. In theory, the high OM concentration from BBOA can facilitate the oxidation product to partition into the aerosol phase (Kroll and Seinfeld, 2008). To investigate the BB influence to IEPOX-SOA: 1) suggest the authors start to see if the BBOA plumes can increase the acidity of aerosol, which can promote the IEPOX uptake (Gaston et al., 2014). 2) the NOx concentration can influence the oxidation product of isoprene (Surratt et al., 2010; Bates et al., 2014). At the absent of IEPOX measurement, the authors can investigate the IEPOX-SOA concentration level or the fC5H6O abundance based on different categories of NOx concentrations by checking if there is systematic differences of IEPOX- SOA mass concentrations at high NOx and low NOx. This analysis presumably assumed the isoprene emissions in the large areas were relatively constant within the field study period.

Response 12: Due to the lack of supporting evidence for either of proposed reasons, this sentence is deleted from the manuscript (Page 17, lines 11-14). 1) According to calculations for the composition depended collection efficiency, 22% of the aerosol were acidic. Furthermore, the correlation of IEPOX-SOA and sulfate is found to be weak (R of 0.3) (Fig. S18). However, two periods (period before 5th of June and after 15th of June) could be clearly separated on the plot. While there is no correlation between sulfate and isoprene-derived OA factor for the first period, when plotting only data collected from the 15th of June, correlation is found to be slightly higher (R of 0.4). The following is added to the manuscript at Page 16 line 6-14: "The main path for

isoprene-derived OA formation is proposed to be acid-catalysed IEPOX uptake (Lambe et al., 2015; Lin et al., 2011). According to calculations for the composition depended collection efficiency, 22% of the aerosol were acidic. In order to estimate whether the acidity of the particles had an influence on isoprene-derived OA generation in ATARS, the correlation between sulfate (taken as proxy of aerosol acidity) and the isoprene-derived OA factor was examined. The correlation between the factor and sulfate can be considered as weak (R of 0.3) (Fig S18). However, two periods can be clearly distinguished from the graph: the period before 5th of June and the period after 15th of June. While there is no correlation between sulfate and the isoprene-derived OA factor for the first period, when plotting only data collected from 15th of June, correlation is found to be slightly higher than the correlation for all background data (R of 0.4)".

2) The comment unfortunately cannot be addressed due to the unreliable NOx data.

Minor comments

Comment 1: Page 4 lines 1-2: Please give the standard deviations of those reported average values.

Response 1: The standard deviation values for all reported average values are given in the revised manuscript. Sentence at Page 4, lines 1-3 has been modified: " The early dry season was characterized by dry weather conditions (average relative humidity of $67 \pm 23$ %) and warm days with an average daily and nightly temperatures of $27 \pm 5$ °C (up to 34 °C) and $19 \pm 4$ °C (with a minimum of 10 °C), respectively".

Comment 2: Page 4, line 27: How can the authors separate aged POA and SOA? To what extent, the authors will define the aerosol is aged POA since SOA can be from aged POA (Robinson et al., 2007).

Response 2: The authors agree that aged POA and SOA cannot be separated using the fragment analysis that was stated by this sentence. Therefore, "aged POA" was excluded from the sentence (Page 4, line 30): "Organic aerosols measured by the AMS

encompassed aerosols that were processed in the atmosphere for different periods of time and included both POA and SOA. As such, a tool was needed to distinguish BB aerosol from other sources and fresh from processed BB aerosol".

Comment 3: Page 5, line 20: Have the authors offset the background of BAM measurement e.g. the background concentration can refer to the mass concentrations of different species on blank filters?

Response 3: Yes, all of the species measured on the BAM PM1 filters were blank corrected. This is added to the manuscript at Page 5, lines 26 and 27: "All of the species measured on BAM filters were blank corrected".

Comment 4: Page 5 line 21: Suggest changing "PM1" to be "PM1", which is more commonly used.

Response 4: "PM1" has been changed in "PM1" throughout the manuscript.

Comment 5: Page 6, line 17: Is the PTR-MS data UMR or HR? How did the authors separate the contributions of isoprene and furan to the m/z 69.

Response 5: An unit mass resolution PTR-MS (quadrupole mass spectrometer) has been used and this is added to the manuscript at Page 6, lines 22-24: "A high sensitivity Proton Transfer Reaction-Mass Spectrometer (PTR-MS, Ionicon Analytik) with a quadrupole mass spectrometer and an H3O+ ion source was employed to measure non-methane organic compounds (NMOCs) that include non-methane hydrocarbons and oxygenated volatile organic compounds".

The authors used the PTR-MS-resolved m/z 69 as a signal related to isoprene. Furan concentrations were significant only at the time of close BB periods (as it can be seen from Fig S17). These have been removed from the data for the PMF (background data) where isoprene-derived OA were examined.

The following is added to the manuscript at Page 7, lines 1 and 2: "In order to distinguish furan and isoprene contribution over the sampling period, a gas chromatographymass spectrometry method was used" and at Page16, lines 15-18: "The isoprene/furan concentrations at m/z 69 (PTR-MS) were treated as an isoprene contribution due to the dominance of isoprene signal compared to furan, according to the samples analysed by gas chromatography-mass spectrometry (Fig. S17). The furan contribution was more significant in BB plumes, during the close BB period, as suggested previously (Warneke et al., 2011)".

Figure (Fig. S17) illustrating the furan and isoprene data analysed by the gas chromatography-mass spectrometry is presented in the supplementary material.

Comment 6: Page 7, line 14: Suggest adding the OM/OC values.

Response 6: The average OM/OC value of 1.4 is stated at Page 7, line 23.

Comment 7: Page 7, line 20: Please change "organic mass" to be OM. Since the authors start to use OM as an abbreviation, it should be consistent in the following text.

Response 7: "OM" has been used as an abbreviation through the manuscript (Page 7, line 23; Page 7, line 24). "Organic mass" is used just at the beginning of the sentence (Page 7, line 27).

Comment 8: Page 7, line 25: Please give the standard deviations to those reported average concentrations.

Response 8: The standard deviation values for all reported average values are given in the new manuscript version at Page 7, line 29 – Page 8, line 2: "In this study, the remaining submicron non-refractory mass was made up of inorganics including sulfates (4.2 %), ammonium (2.8 %), nitrates (1.5 %) and chlorides (1.3 %), with average concentrations of $0.51 \pm 0.32$ $\mu$g m-3, $0.35 \pm 0.68$ $\mu$g m-3, $0.19 \pm 0.45$ $\mu$g m-3 and $0.17 \pm 1.28$ $\mu$g m-3, respectively."

Comment 9: Page 8 line 25: It is not clear that Org 60 is UMR or HR. The authors already use m/z 60 refers to UMR and C2H4O2+ refers to HR. Please be consistent. F60 (HR or UMR) has a similar problem e.g., in page 9 line 1.

[Figure]

Response 9: HR peak fitting data (PIKA) were used through the analysis as stated in Page 4, line 27: "Therefore, HR peak fitting data (PIKA) were chosen for further analysis". Since these are UMR data fitted in PIKA, Org60 and f60 are used in the manuscript instead of $C_2H_4O_2^+$ and $fC_2H_4O_2^+$. The same is done with other fragments.

To make it clear the following is modified to Page 5 lines 1 and 2: "An extensively employed fragments analysis using the AMS-extracted parameters f43 (ions $C_3H_7^+$ and $C_2H_3O^+$), f44 ($CO_2^+$) and f60 ($C_2H_4O_2^+$) was applied here' and at Page 4 lines 7-9: "The AMS parameter f60 and accompanying f73 ($C_3H_5O_2^+$) are widely used as BB emission signatures as they are directly related to levoglucosan-like species, which are a substantial fraction of organics emitted in pyrolysis of cellulose (Alfarra et al., 2007; Simoneit et al., 1999)".

Comment 10: Page 9, line 23: What does "atmospheric processing" refer to?

Response 10: Atmospheric processing refer to aging process which is defined at Page 2, lines 21 and 22 as: fresh particle or gaseous species transformed in the atmosphere through photochemical processing.

Comment 11: Page 11, line 4: Repeated sentence, which was already said in page 7 line 29.

Response 11: This sentence (Page 12, lines 10 and 11) is changed for : "Five single source BB events were analysed here (Fig.2)".

Comment 12: Page 11, line 27: OOA should be SOA, OOA PMF factor has not been introduced here yet.

Response 12: OOA was used here as an oxygenated fraction of OA, as defined in Page 2, line 28 and not as a PMF-resolved factor. Therefore OOA is kept as it is.

Comment 13: Page 11, line 8: "As these plumes were mostly related to sources close to the sampling site (within 2 to 10 km), differences in m/z 44 and 43 can be mostly related to different burning conditions." No evidence was shown to support this sentence.

Response 13: The sentence was removed from the revised manuscript at Page 12, lines 14-16.

For figures

Comment 1: Figure 1. Suggest adding EC fraction in Fig. 1(a) to the total AMS fraction measured in Fig. 1(b), then a full chemical composition of PM1 can be obtained.

Response 1: Mass fractions for inorganic species collected by BAM are added to Figure 1 (a) bar so full chemical composition of PM1 is obtained. Manuscript text is modified, Page 7, lines 19 and 20: "Organic carbon made up 72 % and EC 15 % of the measured PM1 on the BAM filters (Fig. 1a)." Fig. 1 at Page 25 and its caption are also updated.

Comment 2: Figure 3. Has the background of org 60 been subtracted (0.003). If the answer is negative, please offset the background.

Response 2: The background value of 0.003 is related to f60, not to Org 60. The authors did not offset the background value for Org 60. This is not done in other studies as well.

Comment 3: Figure 4. Empirical triangle area was reported in Cubison et al. (2011). Suggest adding right guide line here as well, which can help to compare this study with other studies.

Response 3: We agree that adding the empirical BBOA-related triangle plot in f44 vs. f60 space, introduced by Cubison et al. (2011) will facilitate comparison with other BB studies. The triangle plot is added to plots in Figure 4 but also to Figure 5 and Figure 6 plots, for the same reason. As Figure 4, 5 and 6 are combined in one panel figure (please see the next comment), the new Figure 4 caption is modified: "f44 vs. f60 coloured by date for (a) close and (b) distant BB periods, by $\Delta O3/\Delta CO$ ratio for (c) close and (d) distant BB periods, and by $\Delta OA/\Delta CO$ ratio for (e) close and (f) distant

BB periods. Vertical black lines refer to the f60 background level of 0.003. Red dashed lines refer to the ambient BBOA-related data introduced by Cubison et al. (2011). Note: ozone data from 29th of May until the 1st of June were not available".

Comment 4: Figure 5-6. They are all identical plots. It will be easier to see by combining them together as one panel figure.

Response 4: We agree that combining all f44 vs. f60 plots into the one panel figure gives better insight into the plots. Figure 4 (Page 27) now represents combined figures (previously Figure 4, Figure 5 and Figure 6).

Comment 5: Figure 9 (b): What is the correlation between BBOA with levoglucosan and K+? what is the correlation between IEPOX-SOA with sulfate and isoprene? Does any compound correlate with OOA? Those external tracers should be added on the diurnal variation plot as well. Figure 9 (c): It is better to add the exact fraction values on the bar.

Response 5: The authors have already examined all correlations that have been suggested in this comment. In overall, the correlation of PMF factors was difficult to access due to the "chopped" background dataset that resulted in limited dot points for correlation. The BBOA had a weak correlation with the levoglucosan and nnsK. Moreover, levoglucosan and nssK data are given every 12h, which is the reason why diurnal trend couldn't give useful information. The correlation of isoprene-derived OA and sulfate has found to be weak (R of 0.3) (Please see an answer on the major comment number 12 from Referee 1). There is no correlation between isoprene and isoprene-derived OA, as it was stated and presented in Figure S14. None of parameters have strong trend with the OOA. The exact fraction values for PMF factor contribution are added to the Figure 7 (c).

Comment 6: Figure 10. There are more f44 vs f82 points from IEPOX-SOA factors in Hu et al. (Hu et al., 2015). The authors should add the range of those data, which is more meaningful than showing only two points.

Response 6: The two points present the position of two IEPOX-SOA profiles from two specific studies, Budisulistiorini et al. (2013) and Robinson et al. (2011) in f44 vs. f82 plot.

Alfarra MR, Prevot AS, Szidat S, Sandradewi J, Weimer S, Lanz VA, et al. Identification of the mass spectral signature of organic aerosols from wood burning emissions. Environmental science & technology 2007; 41: 5770-5777. Budisulistiorini SH, Canagaratna MR, Croteau PL, Marth WJ, Baumann K, Edgerton ES, et al. Real-time continuous characterization of secondary organic aerosol derived from isoprene epoxydiols in downtown Atlanta, Georgia, using the Aerodyne Aerosol Chemical Speciation Monitor. Environmental science & technology 2013; 47: 5686-5694. Desservettaz M, Paton-Walsh C, Griffith D, Kettlewell G, Keywood M, Schoot MV, et al. Emission factors of trace gases and particles from tropical savanna fires in Australia. submitted to Journal of Geophysical Research 2016. Mallet MD, Desservettaz MJ, Miljevic B, Milic A, Ristovski ZD, Alroe J, et al. Biomass burning emissions in north Australia during the early dry season: an overview of the 2014 SAFIRED campaign. Phys. Discuss., doi:10.5194/acp-2016-866, in review, 2016 2016. Martins JV, Dunlap MR, Liousse C. Physical, chemical, and optical properties of regional hazes dominated by smoke in Brazil. Journal of Geophysical Research 1998; 103: 32,059-32,080. Robinson N, Hamilton J, Allan J, Langford B, Oram D, Chen Q, et al. Evidence for a significant proportion of Secondary Organic Aerosol from isoprene above a maritime tropical forest. Atmospheric Chemistry and Physics 2011; 11: 1039-1050. Simoneit BR, Schauer JJ, Nolte C, Oros DR, Elias VO, Fraser M, et al. Levoglucosan, a tracer for cellulose in biomass burning and atmospheric particles. Atmospheric Environment 1999; 33: 173-182. Warneke C, Roberts J, Veres P, Gilman J, Kuster W, Burling I, et al. VOC identification and inter-comparison from laboratory biomass burning using PTR-MS and PIT-MS. International Journal of Mass Spectrometry 2011; 303: 6-14.

Please also note the supplement to this comment:
http://www.atmos-chem-phys-discuss.net/acp-2016-730/acp-2016-730-AC2-

supplement.pdf

**Supplement:**

[revised manuscript text omitted]

**Supplementary material**

**S1 Data analysis**

High organic loadings during heavy biomass burning (BB) episodes interfered with sulfate detection by the AMS in unit mass resolution (UMR), resulting in negative sulfate readings and scattered data points (Fig. S1a). Adjustments in Squirrel to

10 the fragmentation table addressed the negative data points (Fig. S1b), however significant data dispersion still remained. High resolution (HR) fitting using PIKA improved correlation among different sulfate fragments (Fig. S1c).

[Figure]

**Figure S1: Sulfate fragments plots (a) before and (b) after Squirrel fragmentation table adjustments, and (c) after HR fitting in PIKA; $SO_4^{2-}$_x indicates sulfate fragments at m/z 64, 80, 81 and 96 plotted against the sulfate fragment at m/z 48.**

PM1 soluble ions measured by ion suppressed chromatography and OM (converted from OC that was determined by the Thermal-Optical Carbon Analyser) were compared to the corresponding AMS UMR and HR data. A considerable improvement was observed in the HR analysis results for sulfate (Fig. S2b), with R changing from 0.4 to 0.8. HR fitting did not result in significant change for the other inorganic species or organics. However, improvement in the sulfate signals was significant and HR peak fitting data were used in further  analysis.

[Figure]

**Figure S2:** Correlation between BAM PM1 soluble ions and corresponding AMS species including (a) chloride, (b) sulfate, (c) organics, (d) ammonia and (e) nitrate; The lighter points and first number present correlation of BAM data with UMR AMS data, while the darker points and the second number illustrates the correlation for BAM and HR AMS data. Red line represents 1:1 line (absolute concentration between AMS and BAM). BAM organic mass (OM) was converted from OC mass using the conversion coefficient of 1.4. R refers to Pearson correlation coefficient. The AMS data were averaged to BAM 12h data.

**Table S1:** Correlation values between inorganic species and organics during the campaign, and close and distant periods separately. Inorganic species measured during the whole period (X), close BB periods (X(c)) and distant BB periods (X(d)) were compared to organics measured during the same time period (Org, Org(c), Org(d)).

| | $Cl^-$ | $Cl^-$ (c) | $Cl^-$ (d) | $NH_4^+$ | $NH_4^+$ (c) | $NH_4^+$ (d) | $NO_3^-$ | $NO_3^-$ (c) | $NO_3^-$ (d) | $SO_4^{2-}$ | $SO_4^{2-}$ (c) | $SO_4^{2-}$ (d) |
|---|---|---|---|---|---|---|---|---|---|---|---|---|
| **Org** | 0.65 | | | 0.92 | | | 0.75 | | | 0.55 | | |
| **Org(c)** | | 0.67 | | | 0.92 | | | 0.72 | | | 0.49 | |
| **Org(d)** | | | 0.47 | | | 0.73 | | | 0.77 | | | 0.48 |

[Figure]

(a) (b)

**Figure S3:** (a) Time series of chlorides collected on the filters (BAM). Dashed lines are defining the periods with oceanic influence, while solid lines illustrate close BB periods. (b) The filter (BAM) data for magnesium ($Mg^{2+}$) and sodium ($Na^+$). Black line illustrates $Mg^{2+}/Na^+$ ratio for the sea salt (0.12).

**S2 Biomass burning aerosols and aging**

[Figure]

(a) (b)

**Figure S4:** f44 vs. f43 coloured by date for (a) close and (b) distant BB periods. PMF-resolved factors are also indicated. The dashed lines represent boundaries for typical ambient aerosol as presented in Ng et al. (2010).

**S2.1 Secondary organic aerosol (SOA) formation**

[Figure]

**Figure S5: Change in ΔOA/ΔCO ratio with aging (represented by f44 values) for close (crosses) and distant (dots) fires coloured by f60.**

**S2.2 Biomass burning events**

[Figure]

**Figure S6: Normalised mass spectra (sum=1) for selected BB events (A (30/05/14, 18:34-19:25), C (30/05/14, 23:41-31/05/14 00:59), E (09/06/14, 19:45-10/06/14 00:32), F (25/06/14, 12:28-16:59) and G (25/06/14, 21:40-26/06/14 03:59) are shown respectively.**

[Figure]

[Figure]

[Figure]

(c)

[Figure]

(d)

[Figure]

(e)

[Figure]

(f)

[Figure]

(g)

[Figure]

(h)

[Figure]

(i)

[Figure]

(j)

**Figure S 7: Backtrajectories for event A, with close hot spots (a) and close and distant hot spots (b), for event C, with close hot spots (c) and close and distant hot spots (d), for event F, with close hot spots (e) and close and distant hot spots (f), for event G, with close hot spots (g) and close and distant hot spots (h) and for event E, with close hot spots (i) and close and distant hot spots (j). The backtrajectories were computed using the HYSPLIT (Hybrid Single-Particle Lagrangian Integrated Trajectory) model. All backtrajectories were run for 24h back in time. Different colours illustrate different starting time.**

[Figure]

**Figure S 8: f60/f73 ratio vs. modified combustion efficiency (MCE) for different events. Each point label indicates name of event and correlation value between f60 and f73.**

**S3 PMF performed on the whole dataset**

[Figure]

**Figure S9: (a) Q/Qexpected (Q refers to the sum of squared scaled residuals over the whole dataset) vs. number of factors, illustrating high error and residual values. (b) Time series of Q/Q expected contribution for 3 and 6-factor solutions where it is clear that higher number of factors does not make the residual structure during BB events significantly lower.**

[Figure]

**Figure S10: Factor profiles and time series for 3, 4 and 5-factor solutions showing BBOA factor splitting, suggesting that plumes are apportioned to different PMF factors.**

**S4 PMF performed on the background dataset**

[Figure]

**Figure S11: m/z 44 vs. m/z 43 coloured by date for the whole dataset and for background periods (inset). Black dashed box in inset graph illustrates cut-offs of 0.15 for m/z 43 and 0.4 for m/z 44 chosen for background data.**

[Figure]

**Figure S12: (a) Q/Qexpected vs. number of factors and (b) time series of Q/Q expected contribution for the 3-factor solution illustrating significantly smaller residuals in the case of the background dataset, compared to the whole dataset.**

[Figure]

5     **Figure S13: Factor profiles and time series for 2, 4 and 5-factor solutions for the background dataset.**

[Figure]

**Figure S14: Time series for 3-factor solution for the background dataset.**

**S4.1 Isoprene-derived- organic aerosol**

[Figure]

(a)

(b)

5   **Figure S15: HR peak fitting at (a) m/z 82 showing the dominance of the C$_5$H$_6$O$^+$ fragment and (b) m/z 40 demonstrating good m/z calibration.**

[Figure]

**Figure S16: Diurnal trend of PMF IEPOX-SOA factor and isoprene/furan concentration measured by PTR-MS.**

[Figure]

**Figure S 17: Time series of isoprene and furan concentrations analysed by the gas chromatography-mass spectrometry. Black lines illustrate close BB periods.**

[Figure]

**Figure S 18: Correlation between of AMS sulfate with the isoprene-derived OA PMF factor.**

---

## Author Comment (AC3) · 13 Dec 2016

Author response to Referee #3

The authors thank the reviewer for comments and suggestions to improve the manuscript. We have addressed each of comments as outlined below. All page and line numbers refer to the revised manuscript (Revised_manuscript_trackChanges.docx) where all changes (track changes) are available (revised supplementary material is included in the revised manuscript pdf file). The link for the revised manuscript is given at the end of this document. If the text is significantly changed only the number of section is given (e.g. "Biomass burning events", Sect. 3.2.4).

Comment: This paper presents an analysis of atmospheric data from northern Aus-

tralia, mainly AMS data impacted by biomass burning. A range of analyses is performed on the data, which gives some insight into the behaviour of the aerosol regarding the biomass burning plumes, ageing and also the formation of SOA from isoprene through the IEPOX route.

None of the results are particularly earth shattering or unexpected, given the pre-existing literature, and there is nothing really new on a process level compared to previous publications. However, there is currently a lack of in situ characterisation work like this in the tropics, so should probably be publishable on that basis. However, the paper is a little rambling and unfocused in paces, with discussions like isoprene SOA detracting from the supposed subject of the paper (ageing of biomass burning emissions), so the general theme of the paper should maybe be better defined.

Response: Significant changes have been made throughout the manuscript and subject of the paper is better defined. The following has been modified or added to the manuscript: • The title of the manuscript is modified. • Separation on "close BB" and "distant BB" periods is clarified. • More analysis and discussion are added for BB events, including backtrajectories and Sentinel hot spot locations. • The suggestions regarding isoprene-derived OA are incorporated in the manuscript. • Discussion regarding the sea salt-chlorides and f44-resolved size distribution is modified and clarified.

Comment 1: Title: The title is probably not appropriate, given that many other scientific phenomena other than the aging of BB plumes (e.g. IEPOX-SOA) are discussed.

Response 1: Beside the "Biomass burning events" and "isoprene-derived OA" sections (Sect 3.2.4 and Sect. 3.3.1, respectively), the accent of the manuscript is on the investigation of the aerosol changes influenced by the aging processes. The limitation of the stationary measurements made extremely difficult to track any plume with certainty, therefore, we removed the "aging" from the title. A new title is: "Fresh and aged aerosol emitted from biomass burning in northern Australia".

Comment 2: Page 8: "close BB" and "distant BB" should be given more specific definitions. "Prominent" is not very descriptive. Response 2: The authors agree that more specific definitions should be given for "close" and "distant" BB periods and clear presentation of data separation should be given.

The measurement period was separated into "close BB" and "distant BB" periods based on organics, carbon monoxide and particle number concentration and their correlation, as well based on the distance of known fires (Sentinel hot spots) from the ATARS.

Firstly, we were looking at the time series of organics, carbon monoxide and particle number concentration (Figure 2, Page 25) in order to identify the periods of significant BB influences, e.g. their high signals (e.g. on 9th and 26th of June) gave us a guide on which days intense/close BB took place. The increases in BB markers (levoglucosan, soluble non-sea salt potassium and Org 60) also showed the same trend (Figure 3, Page 26). After identifying these periods, the next step was to relate these signals to the possible BB events. Therefore, we have mapped all fires (Sentinel hot spots) that have happened during these periods and have confidence level of 50% and more. To make sure that everything is included (e.g. close fires with low intensity) we have gone through every measurement day and mapped all the events that have occurred.

Hot spots detected within 20 km occurred on 30th of May, between 7th and 11th and 25th and 26th of June (Figure 2, Page 25). Taken the distance of 20 km and increased concentrations of organics, carbon monoxide and particle number, these days were taken as "close BB" periods. Knowing that ATARS was constantly affected by the BB air masses the remaining measurement days were taken as "distant BB" periods. In summary, "close BB" periods refer to periods where close fires (within 20 km) have taken the place and correlated increases in carbon monoxide, organics and particle number concentration were observed (Fig. 2), while the rest of the measurement is defined as "distant BB" periods.

In order to clarify the close and distant BB periods the paragraph at Page 8, line 3- Page 8 Line 16 is modified: "The sampling site was constantly impacted by fire emissions with periods of heavy BBs characterized by high aerosol and gas phase concentrations, for instance CO reaching up to ∼104 ppb and organics up to 350 $\mu$g m-3 (Fig. 2). The most intense BB episodes were on the 30th of May, between the 7th and 11th of June, and on the 25th and 26th of June (Fig. 1c) (Mallet et al., 2016). During intense fire periods, organics, CO and particle number concentration showed correlated increases (Fig. 2). Moreover, looking at Sentinel Hot spots during these periods, hot spots were detected within 20 km from the ATARS. Based on this, the dataset was separated into periods of "close BB" (corresponding to high organics, CO and particle number concentration signals and close events (< 20 km)) and "distant BB" (corresponding to less intense organics, CO and particle number concentration signals and distant events (> 20 km)). It is important to emphasize that all periods during the measurement that have not been included in close BB periods have been assigned to distant BB periods, as ATARS was constantly influenced by BBs. The selection does not mean that emissions from distant fires were not present during the close BB periods but that the influence of fires near the measurement station during these periods was dominant. Nine intense BB events were identified from close BB periods as single source emissions (Desservettaz et al., 2016). Five of the nine BB events (Fig. 2) were analysed here (Sect. 3.2.4), due to the instrument not sampling during the remaining events. Most of the events occurred in the afternoon/night time (Table 1)".

In order to indicate distant BB periods the following is added to Fig. 2 caption at Page 25: "The distant BB periods cover all days of the measurement other than days included in close BB periods".

Comment 3: Page 10: Given the dynamic relationship between NOx and Ozone, it might be sensible to look at perturbations in 'potential ozone' (Ox = NO2 + O3) as well. This would remove the NO titration effect.

Response 3: Unfortunately this comment cannot be addressed as the NOx data were

unreliable due to the issue with the instrumentation.

Comment 4: Page 10: The discussion of OA/CO vs time of day is problematic because the total amount of OA in a plume is the product of the total photochemical history of the air mass, not just the time of day that it was measured at. Without a more detailed treatment of the full air mass history, I do not see how any conclusions regarding things like fragmentation can be made.

Response 4: The authors agree that the photochemical history of an air mass is important when discussing the changes in OA in the plume. To emphasize this the following sentence is added to the manuscript at Page 11, lines 9-11: "It must be noted that simply examining $\Delta OA/\Delta CO$ vs time of day is a simplified approach which does not fully take into account the total photochemical history of the air mass". We have added the diurnal trend of $\Delta O3/\Delta CO$ ratio (Figure 7) as a support for the increased photochemical activity of these masses at the time when OA/CO ratio is increased. In addition, diurnal trend of mentioned parameters is examined also for close fires.

Comment 5: Page 10: How is the change in SMPS size distributions consistent with SOA formation? Have you compared the particulate volume concentrations? Care should be exercised because an increase in mode diameter can occur simply through coagulation processes, which require no additional particulate mass formation.

Response 5: The authors agree that coagulation process is another pathway leading to increase in particle size which cannot be excluded. However, coagulation would not result in OA mass increase. The "f44-resolved size distribution" section (Sect. 3.2.3) is however modified and the sentence "SMPS size distributions consistent with SOA formation" is changed.

Comment 6: Page 11: The assumption that f44 corresponds to photochemical activity is problematic. Biomass burning can produce a large amount of primary HULIS, which has a very high f44. Furthermore, while it has been shown that a plume's f44 will increase with time, it is not proven that photochemistry is necessarily responsible,

particularly in the very early stages after emission where repartitioning or 'dark' chemical processes may occur. I would be more guarded and state that the high f44 implies a high level of oxygenation that could be caused by photochemistry.

Response 6: The authors agree with the referee's comment. The following has been modified: Page 13, lines 17-18: "The detected plumes included a considerable portion of oxygenated OA (average f44 value of 0.13, up to 0.18) that could be caused by high daytime photochemical activity".

Comment 7: Page 13: A lower boundary layer height can increase concentrations of primary emissions, but how would it increase IEPOX-SOA?

Response 7: The authors suggest the following: during the night when the boundary level lowers the increase in concentration of gaseous compounds (including low volatility gas-phase isoprene-derived OA) in the volume of air can induce partitioning of gases onto the particles. Therefore, lower boundary layer during the night might create conditions for low volatility isoprene-derived OA partitioning and increase in isoprene-derived OA. This was added to the manuscript at Page 16, lines 27-30 as following: "During the night, the boundary level lowers which increases the concentration of gaseous compounds and can induce partitioning of gases onto the particles. Therefore, the lower night-time boundary layer might create conditions for low volatility isoprene-derived OA partitioning and an increase in isoprene-derived OA".

Comment 8: Page 14: There may be other explanations for a different f82, such as the formation of isoprene SOA through other routes (e.g. MPAN).

Response 8: The authors agree that other pathways for isoprene-derived SOA formation exist and can contribute to f82 signal. Therefore the "IEPOX-SOA" factor has been renamed to "isoprene-derived OA" (throughout the manuscript), to include the possibility of other pathways in OA formation from isoprene and contribution to f82 through these pathways.

Please also note the supplement to this comment:
http://www.atmos-chem-phys-discuss.net/acp-2016-730/acp-2016-730-AC3-
supplement.pdf

[Figure]

**Supplement:**

[revised manuscript text omitted]

**Supplementary material**

**S1 Data analysis**

High organic loadings during heavy biomass burning (BB) episodes interfered with sulfate detection by the AMS in unit mass resolution (UMR), resulting in negative sulfate readings and scattered data points (Fig. S1a). Adjustments in Squirrel to

10 the fragmentation table addressed the negative data points (Fig. S1b), however significant data dispersion still remained. High resolution (HR) fitting using PIKA improved correlation among different sulfate fragments (Fig. S1c).

[Figure]

**Figure S1: Sulfate fragments plots (a) before and (b) after Squirrel fragmentation table adjustments, and (c) after HR fitting in PIKA; $SO_4^{2-}$_x indicates sulfate fragments at m/z 64, 80, 81 and 96 plotted against the sulfate fragment at m/z 48.**

PM1 soluble ions measured by ion suppressed chromatography and OM (converted from OC that was determined by the Thermal-Optical Carbon Analyser) were compared to the corresponding AMS UMR and HR data. A considerable improvement was observed in the HR analysis results for sulfate (Fig. S2b), with R changing from 0.4 to 0.8. HR fitting did not result in significant change for the other inorganic species or organics. However, improvement in the sulfate signals was significant and HR peak fitting data were used in further  analysis.

[Figure]

**Figure S2:** Correlation between BAM PM1 soluble ions and corresponding AMS species including (a) chloride, (b) sulfate, (c) organics, (d) ammonia and (e) nitrate; The lighter points and first number present correlation of BAM data with UMR AMS data, while the darker points and the second number illustrates the correlation for BAM and HR AMS data. Red line represents 1:1 line (absolute concentration between AMS and BAM). BAM organic mass (OM) was converted from OC mass using the conversion coefficient of 1.4. R refers to Pearson correlation coefficient. The AMS data were averaged to BAM 12h data.

**Table S1:** Correlation values between inorganic species and organics during the campaign, and close and distant periods separately. Inorganic species measured during the whole period (X), close BB periods (X(c)) and distant BB periods (X(d)) were compared to organics measured during the same time period (Org, Org(c), Org(d)).

| | $Cl^-$ | $Cl^-$ (c) | $Cl^-$ (d) | $NH_4^+$ | $NH_4^+$ (c) | $NH_4^+$ (d) | $NO_3^-$ | $NO_3^-$ (c) | $NO_3^-$ (d) | $SO_4^{2-}$ | $SO_4^{2-}$ (c) | $SO_4^{2-}$ (d) |
|---|---|---|---|---|---|---|---|---|---|---|---|---|
| **Org** | 0.65 | | | 0.92 | | | 0.75 | | | 0.55 | | |
| **Org(c)** | | 0.67 | | | 0.92 | | | 0.72 | | | 0.49 | |
| **Org(d)** | | | 0.47 | | | 0.73 | | | 0.77 | | | 0.48 |

[Figure]

(a) (b)

**Figure S3:** (a) Time series of chlorides collected on the filters (BAM). Dashed lines are defining the periods with oceanic influence, while solid lines illustrate close BB periods. (b) The filter (BAM) data for magnesium ($Mg^{2+}$) and sodium ($Na^+$). Black line illustrates $Mg^{2+}/Na^+$ ratio for the sea salt (0.12).

**S2 Biomass burning aerosols and aging**

[Figure]

(a) (b)

**Figure S4:** f44 vs. f43 coloured by date for (a) close and (b) distant BB periods. PMF-resolved factors are also indicated. The dashed lines represent boundaries for typical ambient aerosol as presented in Ng et al. (2010).

**S2.1 Secondary organic aerosol (SOA) formation**

[Figure]

**Figure S5: Change in ΔOA/ΔCO ratio with aging (represented by f44 values) for close (crosses) and distant (dots) fires coloured by f60.**

**S2.2 Biomass burning events**

[Figure]

**Figure S6: Normalised mass spectra (sum=1) for selected BB events (A (30/05/14, 18:34-19:25), C (30/05/14, 23:41-31/05/14 00:59), E (09/06/14, 19:45-10/06/14 00:32), F (25/06/14, 12:28-16:59) and G (25/06/14, 21:40-26/06/14 03:59) are shown respectively.**

[Figure]

[Figure]

[Figure]

(c)

[Figure]

(d)

[Figure]

(e)

[Figure]

(f)

[Figure]

(g)

[Figure]

(h)

[Figure]

(i)

[Figure]

(j)

**Figure S 7: Backtrajectories for event A, with close hot spots (a) and close and distant hot spots (b), for event C, with close hot spots (c) and close and distant hot spots (d), for event F, with close hot spots (e) and close and distant hot spots (f), for event G, with close hot spots (g) and close and distant hot spots (h) and for event E, with close hot spots (i) and close and distant hot spots (j). The backtrajectories were computed using the HYSPLIT (Hybrid Single-Particle Lagrangian Integrated Trajectory) model. All backtrajectories were run for 24h back in time. Different colours illustrate different starting time.**

[Figure]

**Figure S 8: f60/f73 ratio vs. modified combustion efficiency (MCE) for different events. Each point label indicates name of event and correlation value between f60 and f73.**

**S3 PMF performed on the whole dataset**

[Figure]

**Figure S9: (a) Q/Qexpected (Q refers to the sum of squared scaled residuals over the whole dataset) vs. number of factors, illustrating high error and residual values. (b) Time series of Q/Q expected contribution for 3 and 6-factor solutions where it is clear that higher number of factors does not make the residual structure during BB events significantly lower.**

[Figure]

**Figure S10: Factor profiles and time series for 3, 4 and 5-factor solutions showing BBOA factor splitting, suggesting that plumes are apportioned to different PMF factors.**

**S4 PMF performed on the background dataset**

[Figure]

**Figure S11: m/z 44 vs. m/z 43 coloured by date for the whole dataset and for background periods (inset). Black dashed box in inset graph illustrates cut-offs of 0.15 for m/z 43 and 0.4 for m/z 44 chosen for background data.**

[Figure]

**Figure S12: (a) Q/Qexpected vs. number of factors and (b) time series of Q/Q expected contribution for the 3-factor solution illustrating significantly smaller residuals in the case of the background dataset, compared to the whole dataset.**

[Figure]

5     **Figure S13: Factor profiles and time series for 2, 4 and 5-factor solutions for the background dataset.**

[Figure]

**Figure S14: Time series for 3-factor solution for the background dataset.**

**S4.1 Isoprene-derived- organic aerosol**

[Figure]

(a)

(b)

5   **Figure S15: HR peak fitting at (a) m/z 82 showing the dominance of the C$_5$H$_6$O$^+$ fragment and (b) m/z 40 demonstrating good m/z calibration.**

[Figure]

**Figure S16: Diurnal trend of PMF IEPOX-SOA factor and isoprene/furan concentration measured by PTR-MS.**

[Figure]

**Figure S 17: Time series of isoprene and furan concentrations analysed by the gas chromatography-mass spectrometry. Black lines illustrate close BB periods.**

[Figure]

**Figure S 18: Correlation between of AMS sulfate with the isoprene-derived OA PMF factor.**

---

## Referee Report (RR1)

Review of "Fresh and aged aerosols emitted from biomass burning in northern Australia" by Milic et al.

I thank the authors for taking time to revise the manuscript. The authors have addressed the comments adequately. However, I have one minor comment. In response#2 to reviewer#1 (and corresponding places in the revised manuscript), please remove or rephrase the sentence "In order to estimate whether the acidity of the particles had an influence on isoprene-derived OA generation in ATARS, the correlation between sulfate (taken as proxy of aerosol acidity) and isoprene-derived OA factor was examined." This is because sulfate influences isoprene-derived OA through multiple ways (particle acidity, volume, surface area, liquid water, etc) and sulfate cannot be taken as proxy of aerosol acidity.

---

## Referee Report (RR2)

This paper has been greatly improved after the first revision. After reading through the revised paper and response, a minor revision was considered here:

Major comments:

1) To response 6:

The addressing here is quite right. The dilution will not change the f44/f60 ratios assuming the variation of volatility due to dilution will not change the SOA/BBOA fraction very much. However, the mixing with other plumes which contains comparable or higher OA mass concentration compare to BB plumes, will greatly change the f44/f60 ratio. E.g., If the biomass burning plumes mixed with the biogenic dominated air mass, f60 will decrease and f44 is possibly going to change as well.

Although the authors added the possibility of dilution effect in the explanation, the sentence is not in a proper position (originally in last sentence in the first paragraph of page 10), should be mentioned in the f44 vs f60 explanation of Page 9 line 19 to 21 and other relevant places.

2) For the whole 3.2.2 section.

I did not see the meaning of this paragraph. The ΔOA/ΔCO cannot be used for the SOA formation since the ΔOA/ΔCO (ΔOA or ΔCO) from different fresh BB plumes can vary within a wide range (more than a factor of 10) depending on combustion material, condition etc. E.g., (Aiken et al., 2009). The authors do not know ΔOA/ΔCO from different fresh BB plumes surround this observation site are constant or not. And mixing with fresh or aging plumes from other fires or plumes is possible, which will change the initial ΔOA/ΔCO ratio. I suggest to delete this part.

Similar comment also applies to 3.2.1 section ΔO3/ΔCO. The initial ΔO3/ΔCO is unknown or the authors should give the range of ΔO3/ΔCO for fresh BB plume.

3) Section 3.2.4 (the biomass burning events) is too long. There is too much unnecessary detailed information from each fire. Please shorten this section (cut or move some information to the supplementary materials) and give the necessary conclusion. I did not see the scientific points that the authors want to address here.

Minor comments:

The definition of organics is very wide, which can also refer to gas-phase organics. Please use the abbreviation "OA" instead of "organics" when referring to organic aerosol to avoid confusion.

Page5 Line 16: add abbreviation name of "(OOA)" after "oxygenated OA"

Page 16 line 20 not only "the influence of BB emissions", but also "the aged SOA from different sources." Can influence this f82 ratio.

Aiken, A. C., Salcedo, D., Cubison, M. J., Huffman, J. A., DeCarlo, P. F., Ulbrich, I. M., Docherty, K. S., Sueper, D., Kimmel, J. R., Worsnop, D. R., Trimborn, A., Northway, M., Stone, E. A., Schauer, J. J., Volkamer, R. M., Fortner, E., de Foy, B., Wang, J., Laskin, A., Shutthanandan, V., Zheng, J., Zhang, R., Gaffney, J., Marley, N. A., Paredes-Miranda, G., Arnott, W. P., Molina, L. T., Sosa, G., and Jimenez, J. L. (2009). Mexico City aerosol analysis during MILAGRO using high resolution aerosol mass spectrometry at the urban supersite (T0) - Part 1: Fine particle composition and organic source apportionment, Atmos Chem Phys. 9, 6633-6653

---

## Author Response (AR2)

**Manuscript: acp-2016-730**

Editor comment:

I received three evaluations of your revisions. Overall, all three reviewers feel that the paper can be further enhanced and that the replies to the reviews should be improved. I urge you to put a substantial effort in enhancing the paper and answer to each of the points that in the Reviewers' comments thoroughly.

*Response:*
*The authors thank the reviewers for interest in our work and for the helpful comments to improve the manuscript. We have addressed and incorporated the points raised and we appreciate the invitation by the editor to resubmit the article following revision.*

*Please note that all page and line numbers refer to the revised manuscript and the revised supplementary material with track changes. Author's responses are in italics and text colored in red refers to the added text in the manuscript.*

**Author response to Referee #2**

This paper has been greatly improved after the first revision. After reading through the revised paper and response, a minor revision was considered here.

Major comments:

Comment 1:
To response 6: The addressing here is quite right. The dilution will not change the f44/f60 ratios assuming the variation of volatility due to dilution will not change the SOA/BBOA fraction very much. However, the mixing with other plumes which contains comparable or higher OA mass concentration compare to BB plumes, will greatly change the f44/f60 ratio. E.g., If the biomass burning plumes mixed with the biogenic dominated air mass, f60 will decrease and f44 is possibly going to change as well.

Although the authors added the possibility of dilution effect in the explanation, the sentence is not in a proper position (originally in last sentence in the first paragraph of page 10), should be mentioned in the f44 vs f60 explanation of Page 9 line 19 to 21 and other relevant places.

*Response 1:*
*The authors agree with the referee's comment. The sentence adding the possibility of dilution effect (in more clear form): "One more factor that can influence* the f44 vs f60 trend for both close and distant fire is *mixing of the BB plume with air masses containing aerosol particles from other sources (dilution effect)" is relocated from Page 10, line 7 to Page 9, lines 21 and 22.*

Comment 2:
a) For the whole 3.2.2 section. I did not see the meaning of this paragraph. The ΔOA/ΔCO cannot be used for the SOA formation since the ΔOA/ΔCO (ΔOA or ΔCO) from different fresh BB plumes can vary within a wide range (more than a factor of 10) depending on combustion material, condition etc. E.g., (Aiken et al., 2009). The authors do not know ΔOA/ΔCO from different fresh BB plumes surround this observation site are constant or not. And mixing with fresh or aging plumes from other fires or plumes is possible, which will change the initial ΔOA/ΔCO ratio. I suggest to delete this part.

b) Similar comment also applies to 3.2.1 section $\Delta O_3/\Delta CO$. The initial $\Delta O_3/\Delta CO$ is unknown or the authors should give the range of $\Delta O_3/\Delta CO$ for fresh BB plume.

*Response 2:*
*a) The authors agree that not enough data are available for the conclusions regarding the SOA formation (ΔOA/ΔCO diurnal trend analysis). Therefore, the authors removed this part (Page 11, line 1 to Page 11 line 12) from the revised manuscript, as suggested. Figure 5 that corresponds to the removed text was also removed from the revised manuscript version (Page 26). The first part of the 3.2.2 section which says "Since increased photochemical activity was identified in BB air masses, the change in ΔOA/ΔCO ratio was investigated in order*

to determine whether additional OA was produced in the BB plumes (Fig. 4e and Fig. 4f). The lowest OA concentration observed during the campaign of 0.09 μg m$^{-3}$ was taken as a background value in these calculations. Figures show no particular trend in ΔOA/ΔCO ratio with f44. The ΔOA/ΔCO ratio remains quite constant despite increases in f44 (Fig. S5)" was moved to the end of the previous section 3.2.1 (Ozone formation). This part illustrates that there is no particular trend observed, considering all sampled plumes on whether the aging leads to additional OA generation, due to number of different plume masses.

b) Although the initial value for the ΔO$_3$/ΔCO is unknown, here the authors are considering the overall trends, more precisely ozone enrichments with aging (f44 increase) for close and distant emissions and comparison of the average ΔO$_3$/ΔCO ratio between close and distant fires. The trends are clearly showing higher photochemical activity with more aged masses. Therefore the paragraph is kept in the manuscript.

Comment 3:
Section 3.2.4 (the biomass burning events) is too long. There is too much unnecessary detailed information from each fire. Please shorten this section (cut or move some information to the supplementary materials) and give the necessary conclusion. I did not see the scientific points that the authors want to address here.

*Response 3:*
*The authors agree and the Section 3.2.4 is shortened. The details about the sentinel hotspots detected on days of biomass burning events, including fire confidence level and location, are moved to the supplementary information (Page 4 and 5 within the Sect S2.2). Some corrections were made throughout the section and more conclusions are made at Page 14, lines 4-13:"In summary, the backtrajectory analysis showed that not more than half an hour was needed for the air masses to reach the ATARS from each area affected by close fire events. All sampled aerosols had f44 values lower than 0.1, which likely reflects fresher aerosol masses. Only aerosols emitted from Event F had higher f44 values, which can be a reflection of aerosol processing but also different factors, including mixing of masses due to the abrupt wind changes. The variation between fires and their burning material and conditions, can also be seen through a wide range of MCE values, from 0.9 to 0.98. Events were characterised with different f60/f73 ratios (varying between 1.2 and 2.5), and there was no trend in the relationship between f60/f73 ratio and burning material/conditions (MCE values) (Fig. S8). In general, the diversity of BB plumes was illustrated through high variability in chemical signature as a product of different burning conditions/material for these five intense fire events". The sentence that emphasise the variability of five fire events and how comprehensive approach and more dataset has to be taken is given in the Conclusion at Page 17, lines 23-25: "Even across these five events, the chemical signature varied significantly (e.g. wide range in f44, from 0.06 to 0.18) which emphasises the need for a more comprehensive dataset that characterises the factors associated with burning material, conditions and processing".*

Minor comments:

Comment 1:
The definition of organics is very wide, which can also refer to gas-phase organics. Please use the abbreviation "OA" instead of "organics" when referring to organic aerosol to avoid confusion.

*Response 1:*
*The abbreviation "OA" is used instead of "organics" when referring to organic aerosol.*

Comment 2:
Page5 Line 16: add abbreviation name of "(OOA)" after "oxygenated OA"

*Response 2: The abbreviation "OOA" is written after the "Oxygenated OA" at Page 5, line 17 of the revised manuscript.*

Comment 3:
Page 16 line 20 not only "the influence of BB emissions", but also "the aged SOA from different sources." Can influence this f82 ratio.

*Response 3:*
*The suggested modification is made to the manuscript at Page 17, lines 2-4: "The lower value for SAFIRED compared to Borneo forest, considering that biogenic influence is significant for both environments, can be attributed to the high influence of BB emissions but also to the aged SOA that are from different sources".*

**Author response to Referee #1**

Comment 1:

I thank the authors for taking time to revise the manuscript. The authors have addressed the comments adequately.

However, I have one minor comment. In response#2 to reviewer#1 (and corresponding places in the revised manuscript), please remove or rephrase the sentence "In order to estimate whether the acidity of the particles had an influence on isoprene-derived OA generation in ATARS, the correlation between sulfate (taken as proxy of aerosol acidity) and isoprene-derived OA factor was examined." This is because sulfate influences isoprene-derived OA through multiple ways (particle acidity, volume, surface area, liquid water, etc) and sulfate cannot be taken as proxy of aerosol acidity.

*Response 1:*

*The authors agree that sulfate can influence isoprene through multiple ways and cannot be taken simply as a proxy of aerosol acidity. Therefore, the paragraph is removed from the manuscript at Page 15 of the revised manuscript and Figure S18 (Page 15 of the supplementary information) that corresponds to the paragraph was removed as well.*

**Author response to Referee #3**

Comment 1: While I thank the authors for revising the title, it is still exclusively biomass burning focused, which does not reflect the content of the paper.

*Response 1: Authors agree with the comment. Title has been rephrased for "Biomass burning and biogenic aerosols in northern Australia during the SAFIRED campaign".*

Comment 2: The response regarding the increase in isoprene SOA in response to the boundary layer height is not adquate. For isoprene to build up during a nocturnal inversion it must be continuously emitted, however biogenic isoprene is generally only emitted during the day. I would expect that a different mechanism is reponsible, such as a change in wind sector or partitioning of semivolatile species, but the PTR-MS should indicate one way or another whether an increase in precursor concentrations is responsible.

*Response 2: The authors think that the isoprene doesn't have to be necessarily emitted continuously for isoprene oxidation products to partition over the night. The lower boundary layer is "concentrating" the gases inducing the partitioning of gaseous compounds including the low volatility isoprene-derived OA. To make this more clear, the paragraph (Page 16, lines 14-20) is modified:* *"This night-time enhancement might be due to transport of distant air masses, or partitioning of lower volatility species into the particle phase as the temperature drops and relative humidity increases, as suggested by Budisulistiorini et al. (2013). In addition, the boundary layer height becomes lower during the night, increasing the concentration of gaseous compounds. This boundary layer reduction might induce low volatility OA partitioning, contributing to the increase in isoprene-derived OA".*

Comment 3: Regarding the dOA/dCO analysis vs time of day, it would be useful to state the estimated plume ages here, based on HYSPLIT and the MODIS data. This would give an indication of how meaningful the diurnal profiles are likely to be.

*Response 3: The part regarding the SOA formation (ΔOA/ΔCO analysis) is removed from the manuscript (Page 11, line 1 to Page 11 line 12), as suggested by the Referee #2 and due to insufficient data for further*

*conclusions. Figure 5 that corresponds to the removed text was also removed from the revised manuscript version (Page 26). The first part of the section which says "Since increased photochemical activity was identified in BB air masses, the change in ΔOA/ΔCO ratio was investigated in order to determine whether additional OA was produced in the BB plumes (Fig. 4e and Fig. 4f). The lowest OA concentration observed during the campaign of 0.09 µg m-3 was taken as a background value in these calculations. Figures show no particular trend in ΔOA/ΔCO ratio with f44. The ΔOA/ΔCO ratio remains quite constant despite increases in f44 (Fig. S5)"was moved to the end of the previous section 3.2.1 (Ozone formation). This part illustrates that there is no particular trend observed considering all sampled plumes on whether the aging leads to additional OA generation, due to number of different plume masses.*

[revised manuscript text omitted]

**S1 Data analysis**

High organic aerosol (OA) loadings during heavy biomass burning (BB) episodes interfered with sulfate detection by the AMS in unit mass resolution (UMR), resulting in negative sulfate readings and scattered data points (Fig. S1a). Adjustments in Squirrel to the fragmentation table addressed the negative data points (Fig. S1b), however significant data dispersion still remained. High resolution (HR) fitting using PIKA improved correlation among different sulfate fragments (Fig. S1c).

[Figure]

**Figure S1: Sulfate fragments plots (a) before and (b) after Squirrel fragmentation table adjustments, and (c) after HR fitting in PIKA; SO$_4^{2-}$_x indicates sulfate fragments at m/z 64, 80, 81 and 96 plotted against the sulfate fragment at m/z 48.**

PM1 soluble ions measured by ion suppressed chromatography and OM (converted from OC that was determined by the Thermal-Optical Carbon Analyser) were compared to the corresponding AMS UMR and HR data. A considerable improvement was observed in the HR analysis results for sulfate (Fig. S2b), with R changing from 0.4 to 0.8. HR fitting did not result in significant change for the other inorganic species or organics. However, improvement in the sulfate signals was significant and HR peak fitting data were used in further analysis.

[Figure]

**Figure S2:** Correlation between BAM PM1 soluble ions and corresponding AMS species including (a) chloride, (b) sulfate, (c) OA, (d) ammonia and (e) nitrate; The lighter points and first number present correlation of BAM data with UMR AMS data, while the darker points and the second number illustrates the correlation for BAM and HR AMS data. Red line represents 1:1 line (absolute concentration between AMS and BAM). BAM organic mass (OM) was converted from OC mass using the conversion coefficient of 1.4. R refers to Pearson correlation coefficient. The AMS data were averaged to BAM 12h data.

**Table S1:** Correlation values between inorganic species and  OA during the campaign, and close and distant periods separately. Inorganic species measured during the whole period (X), close BB periods (X(c)) and distant BB periods (X(d)) were compared to  OA measured during the same time period (Org, Org(c), Org(d)).

| | $Cl^-$ | $Cl^-$ (c) | $Cl^-$ (d) | $NH_4^+$ | $NH_4^+$ (c) | $NH_4^+$ (d) | $NO_3^-$ | $NO_3^-$ (c) | $NO_3^-$ (d) | $SO_4^{2-}$ | $SO_4^{2-}$ (c) | $SO_4^{2-}$ (d) |
|---|---|---|---|---|---|---|---|---|---|---|---|---|
| **Org** | 0.65 | | | 0.92 | | | 0.75 | | | 0.55 | | |
| **Org(c)** | | 0.67 | | | 0.92 | | | 0.72 | | | 0.49 | |
| **Org(d)** | | | 0.47 | | | 0.73 | | | 0.77 | | | 0.48 |

[Figure]

**Figure S3:** (a) Time series of chlorides collected on the filters (BAM). Dashed lines are defining the periods with oceanic influence, while solid lines illustrate close BB periods. (b) The filter (BAM) data for magnesium ($Mg^{2+}$) and sodium ($Na^+$). Black line illustrates $Mg^{2+}/Na^+$ ratio for the sea salt (0.12).

**S2 Biomass burning aerosols and aging**

[Figure]

**Figure S4: f44 vs. f43 coloured by date for (a) close and (b) distant BB periods. PMF-resolved factors are also indicated. The dashed lines represent boundaries for typical ambient aerosol as presented in Ng et al. (2010).**

5   **S2.1 Secondary organic aerosol (SOA) formation**

[Figure]

**Figure S5: Change in ΔOA/ΔCO ratio with aging (represented by f44 values) for close (crosses) and distant (dots) fires coloured by f60.**

**S2.2 Biomass burning events**

[Figure]

**Figure S6: Normalised mass spectra (sum=1) for selected BB events (A (30/05/14, 18:34-19:25), C (30/05/14, 23:41-31/05/14 00:59), E (09/06/14, 19:45-10/06/14 00:32), F (25/06/14, 12:28-16:59) and G (25/06/14, 21:40-26/06/14 03:59) are shown respectively.**

**Sentinel hotspots and backtrajectories**

On 30[th] of May at around 2pm three hot spots (two having confidence level of approximately 50% and one of 70%) were detected within 2km on the NE from the ATARS (Fig. S7a - Fig.S7d). These hot spots likely illustrated two fire events. On the same day and time, 11km on the SE from the sampling site, cluster of events was observed, including 4 hot spots with the confidence level between 94% and 100% and one of 78% confidence level. As all of them were spotted at the same time and within 1km distance, it is most likely that the one big fire has occurred. No other close events were observed over this time

period. Cluster of hot spots was detected on the SE approximately 50km from the ATARS and big clusters at 100km and 150km, as well as on the SE. Moreover, 200km on E along the backtrajectories cluster of hot spots was observed.

On 25$^{th}$ of June three hotspot clusters were observed close to ATARS (2km on E, 5km on NE and 10km on SE) (Fig. S7e-Fig. S7h). The cluster of hot spots observed 10 km from the sampling site had one of the highest hot spot's power (energy released by the fire) observed close to the ATARS (within 20 km) during the campaign (120 MW/km$^2$). Besides the close fires two big clusters around 60 km and 120 km on the SE from ATARS were detected on the same day.

On the 9$^{th}$ of June cluster of hot spots (all with the confidence level higher than 70%) was detected 5 km from ATARS. Number of distant hot spots was detected between 100 km and 200 km, on the SE from the ATARS.

Backtrajectories for all BB Events are given at Fig. S7.

[Figure]

(a)

[Figure]

(b)

[Figure]

(c)

[Figure]

(d)

[Figure]

(e)

[Figure]

(f)

[Figure]

(g)

[Figure]

(h)

[Figure]

(i)

[Figure]

(j)

**Figure S 7: Backtrajectories for  Event A, with close hot spots (a) and close and distant hot spots (b), for  Event C, with close hot spots (c) and close and distant hot spots (d), for  Event F, with close hot spots (e) and close and distant hot spots (f), for  Event G, with close hot spots (g) and close and distant hot spots (h) and for  Event E, with close hot spots (i) and close and distant hot spots (j). The backtrajectories were computed using the HYSPLIT (Hybrid Single-Particle Lagrangian Integrated Trajectory) model. All backtrajectories were run for 24h back in time. Different colours illustrate different starting time. (Google Earth V 7.1.2.2041; December 5, 2016; Northern Territory, Australia).**

[Figure]

[Figure]

**Figure S 8: f60/f73 ratio vs. modified combustion efficiency (MCE) for different events. Each point label indicates name of event and correlation value between f60 and f73.**

**S3 PMF performed on the whole dataset**

[Figure]

Figure S9: (a) Q/Qexpected (Q refers to the sum of squared scaled residuals over the whole dataset) vs. number of factors, illustrating high error and residual values. (b) Time series of Q/Q expected contribution for 3 and 6-factor solutions where it is clear that higher number of factors does not make the residual structure during BB events significantly lower.

[Figure]

**Figure S10: Factor profiles and time series for 3, 4 and 5-factor solutions showing BBOA factor splitting, suggesting that plumes are apportioned to different PMF factors.**

**S4 PMF performed on the background dataset**

[Figure]

**Figure S11: m/z 44 vs. m/z 43 coloured by date for the whole dataset and for background periods (inset). Black dashed box in inset graph illustrates cut-offs of 0.15 for m/z 43 and 0.4 for m/z 44 chosen for background data.**

[Figure]

**Figure S12:** (a) Q/Qexpected vs. number of factors and (b) time series of Q/Q expected contribution for the 3-factor solution illustrating significantly smaller residuals in the case of the background dataset, compared to the whole dataset.

[Figure]

5    **Figure S13: Factor profiles and time series for 2, 4 and 5-factor solutions for the background dataset.**

[Figure]

**Figure S14: Time series for 3-factor solution for the background dataset.**

**S4.1 Isoprene-derived- organic aerosol**

[Figure]

(a)    (b)

5   **Figure S15: HR peak fitting at (a) m/z 82 showing the dominance of the $C_5H_6O^+$ fragment and (b) m/z 40 demonstrating good m/z calibration.**

[Figure]

**Figure S16: Diurnal trend of PMF IEPOX-SOA factor and isoprene/furan concentration measured by PTR-MS.**

[Figure]

**Figure S 17: Time series of isoprene and furan concentrations analysed by the gas chromatography-mass spectrometry. Black lines illustrate close BB periods.**

[Figure]

5 |